# PARAMETRIC SDF FOR DYNAMIC SURFACE RECONSTRUCTION

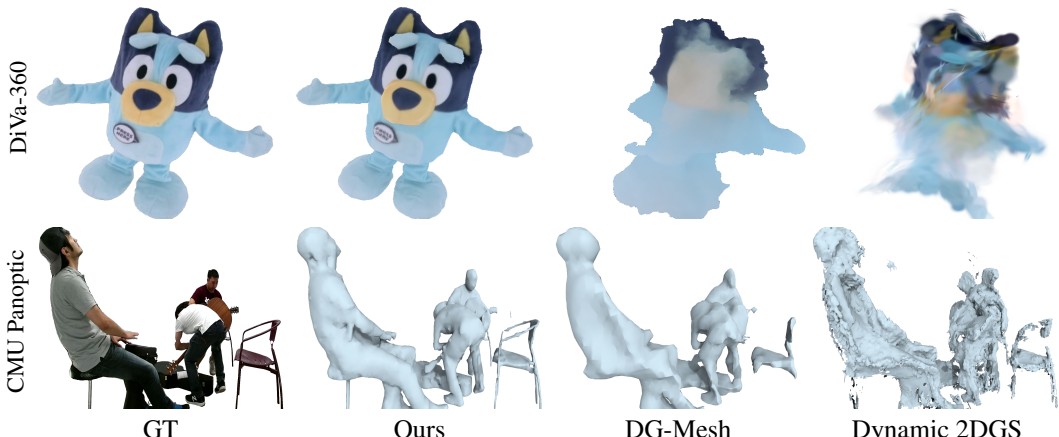

Figure 1: p-SDF produces high-fidelity dynamic 3D surface reconstructions. Compared with prior arts (Liu et al., 2024; Zhang et al., 2024), our method preserves more faithful appearance and geometry across challenging real-world dynamic datasets (Joo et al., 2015; Lu et al., 2024).

## ABSTRACT

Reconstructing high-fidelity, temporally coherent surfaces of dynamic scenes remains a critical challenge in computer vision. While recent methods excel at novel view synthesis, they often fail to recover accurate geometry, yielding noisy or temporally inconsistent meshes that are suboptimal for downstream applications such as simulation or editing. In this work, we introduce a new paradigm for dynamic surface reconstruction based on a parametric Signed Distance Function (p-SDF). Our key insight is to generalize static SDF fields—where each spatial point stores a constant value—into time-dependent parametric curves with each curve modeling a temporally evolving SDF trajectory. Such a parametric SDF modeling provides a principled way to capture complex temporal variations, naturally enforcing smoothness and continuity in shape dynamics. At each timestamp, a static SDF field can be queried from p-SDF and converted into an explicit surface mesh via differentiable iso-surfacing. By rendering these meshes with a physically based differentiable renderer, we optimize the underlying parametric curves end-to-end against 2D image observations. Our framework produces high-fidelity, temporally coherent surfaces and inherently disentangles geometry, material, and lighting from multi-view videos. It robustly reconstructs geometry under large-scale motions and resolves appearance ambiguities caused by challenging lighting and occlusions. Experiments on both synthetic and real-world scenes demonstrate that our method achieves state-of-the-art geometric accuracy and temporal consistency, delivering delicate meshes that surpass prior work, benefiting from our parametric SDF representation.

## 1 INTRODUCTION

Reconstructing high-fidelity surfaces from multi-view video is a long-standing pursuit in computer vision, powering applications in simulation, virtual and augmented reality (VR/AR), autonomous robotics, and film production. A central requirement is *temporal coherence*: the reconstructed geometry must evolve smoothly and consistently over time. Without it, surfaces may flicker, distort,

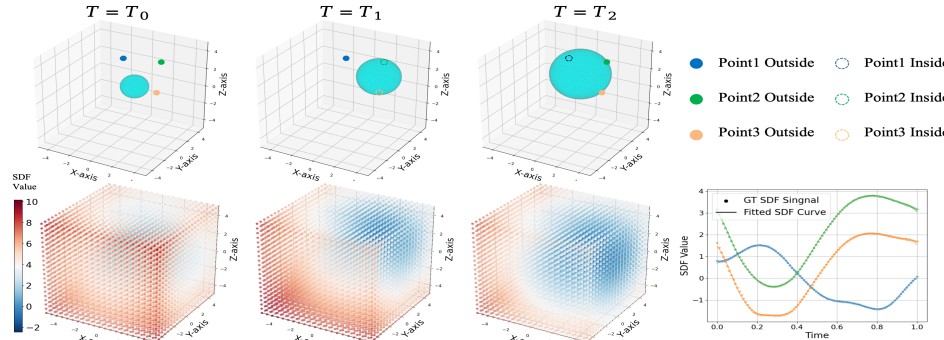

Figure 2: **Continuous SDF Trajectory.** The SDF field of a dynamic scene evolves over time: a dilating and moving sphere (top left) induces corresponding SDF variations (bottom left). Each spatial point, with solid markers for positive (outside) SDF and dashed markers for negative (inside), traces a continuous SDF trajectory. We optimize p-SDF with a compact set of basis functions (Sec. 3.3) to represent these curves. As shown in the figure, p-SDF closely follows the GT SDF trajectories.

or break apart—undermining downstream tasks such as relighting, physical simulation, or editing. Achieving this coherence, however, remains highly challenging, as continuous deformations of dynamic scenes often involve non-rigid motion, large-scale dynamics, and even topological changes.

Recent advances in scene representations, such as Neural Radiance Fields (NeRFs) (Mildenhall et al., 2021) and 3D Gaussian Splatting (3DGS) (Kerbl et al., 2023), have dramatically advanced novel view synthesis (NVS). Their dynamic extensions (Pumarola et al., 2021b; Fridovich-Keil et al., 2023; Cao & Johnson, 2023; Yang et al., 2024b; Duan et al., 2024) further enable photorealistic NVS across time. Nevertheless, these methods are designed primarily for appearance modeling and thus struggle with geometry, often producing noisy or temporally unstable surfaces. Two lines of work have explored improving quality for dynamic surface reconstruction. The first strategy learns a deformation field that maps a canonical template to the dynamic scene over time (Liu et al., 2024; Zhang et al., 2024). While this strategy enforces temporal consistency, it is inherently limited by the fixed template, which fails to represent topological changes. The second strategy directly models the 4D spatio-temporal volume through factorized representations such as Tri-planes or Hash grids (Wang et al., 2023; Chen et al., 2025; Zheng et al., 2025). Although geometry can be extracted by querying the 4D volume, these methods fail to maintain temporal coherence, often producing noisy or flickering reconstructions under complex dynamics or large motions.

In this work, we propose parametric Signed Distance Function (p-SDF), a novel dynamic scene representation that reconstructs high-fidelity, temporally coherent surfaces across time. Our key intuition is that most shape deformations are continuous along time, and the evolution of SDF values at a particular 3D location follows a continuous trajectory (as shown in a toy example in Fig. 2, more detailed analysis in Appendix A). This observation motivates us to generalize static SDF fields, where each spatial point stores a constant scalar value, into time-dependent parametric curves, where temporally evolving SDF trajectory is parametrized with a curve defines. By using a compact set of basis functions to represent these parametric curves, our p-SDF naturally enforces the temporal coherence of the geometry while being flexible in capturing complex dynamics, including large deformations and topological changes.

Specifically, we represent the dynamic geometry with a grid where each vertex stores the parameters of the basis functions. At any timestamp, we can directly query the curve to obtain a static SDF field, from which an explicit mesh can be extracted by differentiable iso-surfacing (Shen et al., 2023). The scene's appearance is modeled separately with an efficient 4D hash grid that predicts physically based (PBR) material properties. The resulting mesh and materials are then passed to a differentiable renderer, allowing the entire framework to be optimized end-to-end against 2D image observations. Our method offers two key advantages: **(i)** p-SDF directly models the variations of SDF fields, enforcing temporal coherence for geometry while allowing topological changes and large-scale motions; **(ii)** by disentangling geometry from a physically based appearance modeling, our framework robustly reconstructs surfaces under challenging lighting and occlusion conditions.

Our p-SDF presents a principled and efficient solution for dynamic 3D reconstruction. We evaluate our method on synthetic and real-world datasets, demonstrating significant performance improvement over prior approaches in geometric accuracy and temporal consistency. Moreover, the high-quality material decomposition from our framework readily supports relighting applications, enabling physically accurate relighting effects.

## 2 RELATED WORK

### 2.1 NEURAL REPRESENTATIONS FOR DYNAMIC SCENES

The advent of Neural Radiance Fields (NeRF) (Mildenhall et al., 2021) revolutionized static novel view synthesis by representing a scene as a continuous volumetric function learned from a collection of 2D images. Extending this paradigm to dynamic scenes involved modeling a 4D space-time function, often by incorporating time-dependent deformations or a time-variant latent code to map observations to a canonical space (Cao & Johnson, 2023; Guo et al., 2023; Park et al., 2021a;b; Pumarola et al., 2021b; Fang et al., 2022). Among these methods, some are applicable to *multi-view* video settings (Cao & Johnson, 2023; Guo et al., 2023), while others are only demonstrated on *monocular* inputs (Park et al., 2021a;b; Pumarola et al., 2021b).

A more recent paradigm shift in neural rendering came with the introduction of 3D Gaussian Splatting (3DGS) (Kerbl et al., 2023), which replaced the implicit volumetric function with a collection of explicit 3D Gaussians. This approach enables state-of-the-art visual quality at real-time rendering speeds and has quickly become the new frontier for dynamic scene modeling. Typically, these methods learn a deformation field that maps Gaussians from a canonical space to the observed space (Huang et al., 2023; Wu et al., 2024; Xu et al., 2024; Yang et al., 2024b; Wang et al., 2025), or extend 3D Gaussians to native 4D Gaussian representation (Yang et al., 2024a; Duan et al., 2024). Most of these Gaussian-based dynamic novel view synthesis (NVS) representations naturally can be applied to *both multi-view and monocular* video inputs. While these methods are effective for NVS, they do not enforce temporal coherence and often yield noisy, incomplete, or temporally inconsistent meshes (Cai et al., 2024; Chen et al., 2025; Zhang et al., 2024).

### 2.2 DYNAMIC SURFACE RECONSTRUCTION

A distinct branch of work explicitly targets surface reconstruction quality, typically relying on Signed Distance Function (SDF) representations and their variants (Wang et al., 2021; Shen et al., 2021; 2023; Munkberg et al., 2022). Their extensions to dynamic reconstruction often adopt deformation field modeling, where a single high-quality canonical template surface is learned and deformed over time (Cai et al., 2022; Johnson et al., 2023; Wang et al., 2024; Yao et al., 2024; Stotko & Klein, 2025). While this formulation yields temporally stable and smooth surfaces, it inherently assumes fixed topology, making it incapable of handling topological changes in many dynamic scenarios.

Alternatively, a body of work directly models the 4D spatio-temporal volume, thereby avoiding topological constraints (Niemeyer et al., 2019; Choe et al., 2023; Mao et al., 2024; Liu et al., 2024; Sang et al., 2025). These methods typically define a unique implicit surface at each timestamp and recover explicit meshes using iso-surfacing techniques (Ju et al., 2002; Chen et al., 2022; Shen et al., 2021; 2023). However, they lack a strong temporal prior: without canonical guidance, each surface is reconstructed with excessive degrees of freedom, often leading to flickering, jittering, or other temporal artifacts that require heavy regularization.

Another line of work reconstructs dynamic surfaces from *multi-view* videos via per-frame or incremental optimization. NeuS2 (Wang et al., 2023) accelerates NeuS-style neural implicit surface learning for *multi-view* reconstruction and extends it to dynamic scenes by incrementally optimizing a separate SDF for each frame with warm-starting. AT-GS Chen et al. (2025) optimize adaptive, temporally consistent Gaussian surfels per frame with optical flow pre-train model's temporal regularization, while GauSTAR (Zheng et al., 2025) binds Gaussians to mesh surfaces and updates them via unbinding and re-meshing to handle topology changes, they are heavily rely on mluti-view depth information. Such frame-wise formulations treat dynamic reconstruction as a discrete sequence of independent static tasks. By isolating the optimization to individual timestamps, these methods lack a unified global temporal model, resulting in redundant storage of per-frame geometries and limited temporal coherence, as smoothness is merely enforced locally rather than intrinsically.

Similar to our method, SDFFlow (Mao et al., 2024) also models the deformation of SDF fields. However, its integral formulation leads to inefficient training and accumulated error, making it impractical for longer sequences and fail to achieve high-quality reconstruction. In contrast, our method directly parameterizes the the SDF trajectories with basis functions, inherently enforcing continuity regularization. The static SDF at a given timestamp can be obtained efficiently by querying the function, without explicit integration that is extremely time-consuming and error-accumulating.

## 3 METHODOLOGY

We now describe our method. The key idea is to generalize static Signed Distance Function (SDF) fields into parametric SDF trajectories. Below, we first review background concepts on SDF and temporal signal modeling in Sec. 3.1. We then provide an overview of the full framework in Sec. 3.2, followed by detailed descriptions of our parametric SDF formulation (Sec. 3.3) and appearance modeling (Sec. 3.4). Lastly, we talk about the implementation details in Sec. 3.5.

### 3.1 BACKGROUND

Our approach integrates principles from explicit surface reconstruction and temporal signal modeling with basis functions. We first provide preliminary knowledges of these two concepts.

**Surface Reconstruction with Signed Distance Functions.** A 3D surface $\partial\Omega$ can be represented as the zero-level set of a Signed Distance Function (SDF). For any spatial point $x \in \mathbb{R}^3$, the SDF value $s(x)$ is the signed distance to $\partial\Omega$, negative inside the enclosed volume $\Omega$ and positive outside:

$$s(x) = \begin{cases} -\inf_{\mathbf{p} \in \partial\Omega} \|x - \mathbf{p}\|_2 & \text{if } x \in \Omega, \\ \inf_{\mathbf{p} \in \partial\Omega} \|x - \mathbf{p}\|_2 & \text{otherwise.} \end{cases} \tag{1}$$

Explicit surface reconstruction methods (Munkberg et al., 2022) define SDF values on a 3D grid and use differentiable iso-surfacing (Shen et al., 2021; 2023) to extract a triangle mesh. The resulting mesh can then be rendered via a differentiable rasterizer (Laine et al., 2020). While highly effective for static scenes, naively applying such pipelines into dynamics, *i.e.*, applying NVDiffRec via per-frame reconstruction strategy, is computationally expensive and does not guarantee temporal consistency, often resulting in flickering geometry and unstable appearance. Instead, our method develops a unified 4D surface representation that preserves the advantages of explicit meshes while enforcing smooth surface evolution over time.

**Temporally Varying Signal Modeling with Basis Functions.** A standard strategy for representing a temporal signal $S(t)$ is to express it as a linear combination of basis functions, $S(t) = \sum_{k=1}^{K} w_k B_k(t)$, where $\{B_k\}_{k=1}^{K}$ are predefined temporal basis and $\{w_k\}_{k=1}^{K}$ are learnable coefficients. With appropriate choices for basis, a small set of coefficients compactly encodes complex temporal behavior while naturally encouraging smoothness.

The common options of basis include Polynomial functions or Fourier basis functions. Specifically, Polynomial basis functions uses a combination of polynomial functions:

$$B_k^p(t) = t^k \tag{2}$$

which captures smooth and low-frequency trends. Alternatively, Fourier basis functions leverages a combinations of periodic functions based on $\sin, \cos$:

$$B_k^f(t) = a_k \cos(\omega_k t) + b_k \sin(\omega_k t), \tag{3}$$

where the $\omega_k$ is the predefined frequency for $k$-th periodic function, and $a_k$ and $b_k$ is the coefficients. The Fourier basis foundations are well-suited for periodic or high-frequency variations as shown in literature (Shi et al., 2014; Tancik et al., 2020; Wang et al., 2022; Rabich et al., 2024; Li et al., 2024).

### 3.2 FRAMEWORK OVERVIEW

**Motivation.** Representing dynamic scenes requires modeling how surfaces evolve over time. As illustrated in Fig. 2, our key observation is that surface motions induces smooth temporal evolution of the SDF values at any fixed spatial location, *i.e.*, $s(x, t)$ traces a continuous 1D signal. These continuous 1D signals can be efficiently parameterized via a collection of basis functions, which not only offers interpretability of the SDF trajectories, but also naturally enforces temporal coherence in the SDF evolution while retaining flexibility for large deformations and topological changes.

**Pipeline.** Fig. 3 illustrates the full system. Given a timestamp $t$, we first obtain per-node SDF values via evaluating basis functions and extracts an explicit mesh using differentiable iso-surfacing (Shen et al., 2023). Concurrently, to model the appearance, we apply a 4D Hash grid and query the PBR attributes for each vertex in the mesh. A physically based differentiable renderer then synthesizes the target views; gradients backpropagate through geometry, appearance, and rendering, enabling end-to-end optimization (Sec. 3.5). This design yields temporally coherent, high-quality meshes and clean disentanglement of geometry, materials, and lighting.

Figure 3: **Pipeline.** For a timestep $t$, p-SDF query the parametric function at each grid node to get the SDF value (Eq. 4). We use FlexiCubes (Shen et al., 2023) to extract the mesh $M_t = (V_t, F_t)$ (Eq. 5), and query the Dynamic Tri-plane Appearance Field (Sec. 3.4) to obtain $(\mathbf{c}, m, r, o)$ (Eq. 7) at point $(\boldsymbol{x}, t)$. A differentiable PBR shader renders images from multi-view cameras, and gradients propagate through all modules for end-to-end training (Sec. 3.5).

## 3.3 PARAMETRIC SIGNED DISTANCE FUNCTION

**Formulation.** We now introduce our dynamic geometry representation, p-SDF, which extends static SDF fields into the temporal domain. Specifically, the scene geometry is defined within a bounded 3D grid $G$ with $N \times N \times N$ nodes $\{\mathbf{p}_i\}_{i=1}^{N^3}$. Unlike static methods that assign a single SDF value to each node, we model the SDF value at node $\mathbf{p}_i$ as a continuous function of time: $s(\mathbf{p}_i, t) : \mathbb{R}^3 \times \mathbb{R} \rightarrow \mathbb{R}$. Following prior work on temporal signal modeling(Lin et al., 2024), we adopt a hybrid basis combining polynomials and Fourier terms to represent $s(\mathbf{p}_i, t)$:

$$s(\mathbf{p}_i, t) = \sum_{n=1}^{N_p} \mu_n t^n + a_0 + \sum_{n=1}^{N_f} \left( a_n \cos\left(\omega_n t\right) + b_n \sin\left(\omega_n t\right) \right), \tag{4}$$

where $N_p$ is the number of polynomials, and $N_f$ is the highest frequency in Fourier function. Collectively, the entire spatio-temporal SDF field is compactly parameterized by $W \in \mathbb{R}^{N^3 \times (N_p + 2N_f + 1)}$, the matrix of all node coefficients.

**Mesh Extraction.** With our time-varying SDF grids, given a timestamp $t$, we can evaluate the basis function to get per-node SDF values. A high-quality, watertight surface mesh is then extracted using differentiable Dual Marching Cubes (Shen et al., 2023):

$$M_t = \texttt{DualMarchingCubes}\left(\{s(\mathbf{p}_i, t)\}_i^{N^3}\right), \qquad M_t := (V_t, F_t), \tag{5}$$

where $V_t$ and $F_t$ denote the mesh vertices and faces at time $t$.

**Discussion.** This parametric design for surface mesh provides several important benefits; **(i)** the basis formulation inherently enforces temporal coherence, while the combination of polynomial and Fourier bases allows capturing both large-scale deformations and fine-grained high-frequency motions, offering robustness in complex dynamic scenes; **(ii)** we model the geometry by storing pernode curve parameters on a grid, which enables efficient and scalable mesh extraction by directly querying the stored function, thus bypassing the computational overhead of per-point MLPs in prior works. **(iii)** by using the explicit mesh for rendering within the differentiable rendering-optimization loop, we apply direct supervision to the geometry. This ensures a consistent and detailed surface, avoiding the noisy artifacts common in purely implicit methods like NeuS (Wang et al., 2023).

## 3.4 APPEARANCE REPRESENTATION

**Formulation.** Having established a continuous representation for dynamic geometry, we now model the time-varying appearances and materials. Unlike geometry, appearance signals such as RGB textures often undergo abrupt changes at fixed spatial locations and therefore cannot be compactly represented by smooth temporal trajectories. To address this, we separately model the appearance with a temporal tri-plane representation, parameterizing the appearance field with three multi-resolution 3D hash grids, $G_{xyt}$, $G_{xzt}$, and $G_{yzt}$, which cover the spatio-temporal domains

Figure 4: **Dynamic Tri-plane Appearance Field.** Left: an input multi-view video (4D sequence). Middle: three multi-resolution spatio-temporal hash grids $G_{xyt}, G_{xzt}, G_{yzt}$ that slice at time $t$ to produce an instantaneous tri-plane. Right: features sampled from the three planes are concatenated and fed into a small MLP decoder that predicts PBR attributes (albedo $c$, occlusion $o$, roughness $r$, and metallic $m$) for shading and differentiable rendering.

$(x, y, t), (x, z, t)$, and $(y, z, t)$. For a surface point $(x, y, z)$ at time $t$, we query these grids in parallel via trilinear interpolation:

$$\mathbf{f}_{xyt} = G_{xyt}(x, y, t), \quad \mathbf{f}_{xzt} = G_{xzt}(x, z, t), \quad \mathbf{f}_{yzt} = G_{yzt}(y, z, t). \tag{6}$$

The resulting feature vectors are concatenated and decoded by a lightweight MLP $\Phi_{\text{app}}$ into physically based material properties:

$$(\mathbf{c}, m, r, o) = \Phi_{\text{app}}(\text{concat}(\mathbf{f}_{xyt}, \mathbf{f}_{xzt}, \mathbf{f}_{yzt})), \tag{7}$$

where $\mathbf{c}$ denotes the diffuse albedo, $m$ the metallic value, $r$ the surface roughness, and $o$ the self-occlusion attribute. At any timestamp $t$, this formulation defines a complete, fine-grained PBR texture field aligned with the mesh $M_t$ extracted from the p-SDF.

**Rendering.** At time $t$, with the extracted mesh $M_t = (V_t, F_t)$ from the geometry module (Sec. 3.3), we first rasterize it and generate a G-buffer of world-space positions $\mathbf{x}$ and surface normals $\mathbf{n}$. For each visible surface point, the Dynamic Tri-plane Appearance Field is queried at $(\mathbf{x}, t)$ to produce material properties $(\mathbf{c}, m, r, o)$ (Eq. 7). These attributes, combined with the mesh geometry and a shared learnable environment map, are passed to a differentiable PBR shader, following Nvdiffrec (Munkberg et al., 2022), to compute the final rendered RGB image. This design enables end-to-end optimization of both geometry and appearance against multi-view video observations.

**Discussion.** Our appearance representation offers two advantages: **(i)** by decomposing the 4D spatio-temporal domain into three 3D hash grids, it reduces computational complexity to directly model 4D appearance; **(ii)** Benefiting from the precise geometry, the simple approach suffices to achieve a clean disentanglement of shape and appearance and produce high-quality material and lighting estimation.

### 3.5 IMPLEMENTATION DETAILS

**Loss Function.** We employ a compact objective that balances photometric reconstruction with geometric and appearance priors. The overall loss is:

$$\mathcal{L}_{\text{total}} = \mathcal{L}_{\text{photo}} + \mathcal{L}_{\text{curve}} + \mathcal{L}_{\text{app}}, \tag{8}$$

where $\mathcal{L}_{\text{photo}}$ enforces consistency between rendered and ground-truth images, $\mathcal{L}_{\text{curve}}$ regularizes the spatial smoothness of SDF trajectories to ensure coherent motion for neighboring vertices, and $\mathcal{L}_{\text{app}}$ regularizes material and lighting. Detailed formulations of these losses are provided in Appendix B.

**Model Details.** We use a grid of resolution $96^3$ to represent our p-SDF, and use a combination of 6 polynomial basis functions and a set of Fourier components, the number of which is chosen per-scene from the range $[18, 100]$ based on motion complexity. For the Dynamic Tri-plane Appearance Field, we adapt the multi-resolution hash grid following Grid4D (Xu et al., 2024), with spatial resolution spanning from 16 to 2048 and temporal resolution from 8 to the full sequence length.

**Training and Optimization.** We build our framework upon the open-source RadianceFieldStudio codebase (Ye et al., 2025) and conduct all experiments on a single NVIDIA RTX 4090 GPU. We train each scene for 5,000 iterations, which takes approximately 1 to 3 hours. We optimize the model using an Adam optimizer (Kingma, 2014) with exponential decay.

## 4 EXPERIMENTS

### 4.1 EXPERIMENTAL SETUP

**Datasets.** To rigorously validate our method, we evaluate on both synthetic and real-world datasets that feature challenging dynamic scenes. Specifically, we select three datasets:

*SynthoMotion-360.* We first curate a *new synthetic benchmark* designed to assess reconstruction under large, non-rigid deformations that existing datasets rarely capture. While existing datasets primarily feature constrained human or object motions, ours introduces a diverse range of subjects, including animals and animated characters, undergoing extreme non-rigid deformations. These challenging scenarios are absent in current benchmarks. In particular, we carefully select 7 animated assets from Objaverse (Deitke et al., 2022; Liang et al., 2024), each with 70–300 frames rendered in Blender from 38 cameras. We use 32 views for training and 6 for testing. Accurate ground-truth meshes are provided for every frame, enabling quantitative evaluation of both novel view synthesis quality and geometry accuracy. We provide overview of our dataset in Appendix I.

*DiVa-360* is a real-world multi-view dataset featuring tabletop dynamic scenes captured under synchronized RGB cameras (Lu et al., 2024). It provides synchronized videos and foreground masks but no ground-truth meshes, so we only evaluate photometric fidelity in this dataset. We use 4 representative sequences, each using 31 cameras for training and 6 held-out views for evaluation.

*CMU Panoptic Studio* is a representative real-world dynamic dataset featuring complex motions (Joo et al., 2015). Following the protocol of SDFFlow (Mao et al., 2024), we focus on challenging multi-person interaction sequences, each containing 24 frames recorded from 10 calibrated cameras. As this dataset lacks ground-truth meshes, we use it for qualitative comparison to demonstrate the quality of geometry. We use all 10 views for training.

**Metrics.** We evaluate the performance from two aspects: *novel-view synthesis quality* and *geometric reconstruction accuracy*. For novel view synthesis quality, we report PSNR, SSIM (Wang et al., 2004), and LPIPS scores (Zhang et al., 2018). For geometric reconstruction quality, we report the Chamfer-L1 distance and the Mean Angular Error (MAE) of surface normals, following the protocol of Verbin et al. (2022)

**Baselines.** We benchmark our method against state-of-the-art baselines from two representative research directions: *dynamic surface reconstruction* and *dynamic novel view synthesis*. For the first one, we compare with two state-of-the-art works: Dynamic-2DGS (Zhang et al., 2024) and DG-Mesh (Liu et al., 2024), which extracts consistent meshes from dynamic 3D Gaussians. We note that while SDFFlow (Mao et al., 2024) is a conceptually related SDF-based approach, it is excluded due to its prohibitive optimization time, requiring several weeks per scene. For the second one, we compare with three state-of-the-art works: Deformable-3DGS (Yang et al., 2024b), SC-GS (Huang et al., 2023) and Grid4D (Xu et al., 2024). Methods that are originally designed for monocular settings have been adapted to multi-view for fair comparisons. This selection allows a comprehensive evaluation of our model's superiority in both geometric accuracy and novel view synthesis quality.

### 4.2 EXPERIMENTAL RESULTS

**Quantitative Results.** We provide quantitative comparisons with all the baselines in Tbl. 1, with detailed results of each scene in Appendix Tbl. A1 and Tbl. A2. Our approach consistently outperforms all the competing methods across all the metrics on both the synthetic *SynthoMotion-360* and the real-world *DiVa-360* datasets, demonstrating the superiority in reconstructing both geometry and visual appearance. This improvement highlights the core advantage of our p-SDF representation, which inherently enforces temporal consistency and is well suited for capturing complex non-rigid dynamics. With p-SDF providing precise and consistent geometry, the dynamic tri-plane appearance field further produces faithful material decomposition, rendering images that have much better visual quality than rendering-focused state-of-the-art methods without requiring specialized optimizations for appearance modeling.

Table 1: Quantitative evaluation of our method compared to previous work on *SynthoMotion-360* and *DiVa-360* datasets. Our method outperforms existing approaches in terms of both visual fidelity and geometric quality. For clarity, all Chamfer distances are scaled by $10^{-3}$.

| Method | SynthoMotion-360 | | | | | DiVa-360 | | |
|---|---|---|---|---|---|---|---|---|
| | PSNR↑ | SSIM↑ | LPIPS↓ | MAE↓ | Chamfer↓ | PSNR↑ | SSIM↑ | LPIPS↓ |
| Deformable-3DGS | 28.897 | 0.956 | 0.077 | 2.842 | 4.535 | 27.280 | 0.956 | 0.068 |
| SC-GS | 29.881 | 0.962 | 0.058 | 2.372 | 3.051 | 25.723 | 0.951 | 0.075 |
| Dynamic-2DGS | 29.489 | 0.959 | 0.062 | 2.461 | 2.850 | 17.513 | 0.910 | 0.187 |
| Grid4D | 30.048 | 0.962 | 0.061 | 2.639 | 4.637 | 27.408 | 0.954 | 0.065 |
| DG-Mesh | 25.719 | 0.936 | 0.081 | 2.794 | 3.760 | 17.734 | 0.919 | 0.183 |
| Ours | **31.653** | **0.974** | **0.027** | **1.332** | **2.220** | **28.076** | **0.957** | **0.049** |

**Qualitative Results.** We present qualitative comparisons in Fig 1, 5, 6 and 7, showing results from *SynthoMotion360*, *CMU Panoptic Studio* and *DiVa-360*, respectively. Additional results are provided in Appendix Fig. A3, A4, A5 and A6. In geometric reconstruction, the surfaces recovered by baselines are severely flawed, suffering from fragmented geometry, incompleteness and noise under real-world scenarios (Fig. 6). In contrast, our method benefits from the precise and stable geometry learned by our p-SDF, delivering complete, smooth, and topologically sound reconstructions, while also faithfully synthesizing fine textures and maintaining temporal coherence even during complex dynamic sequences.

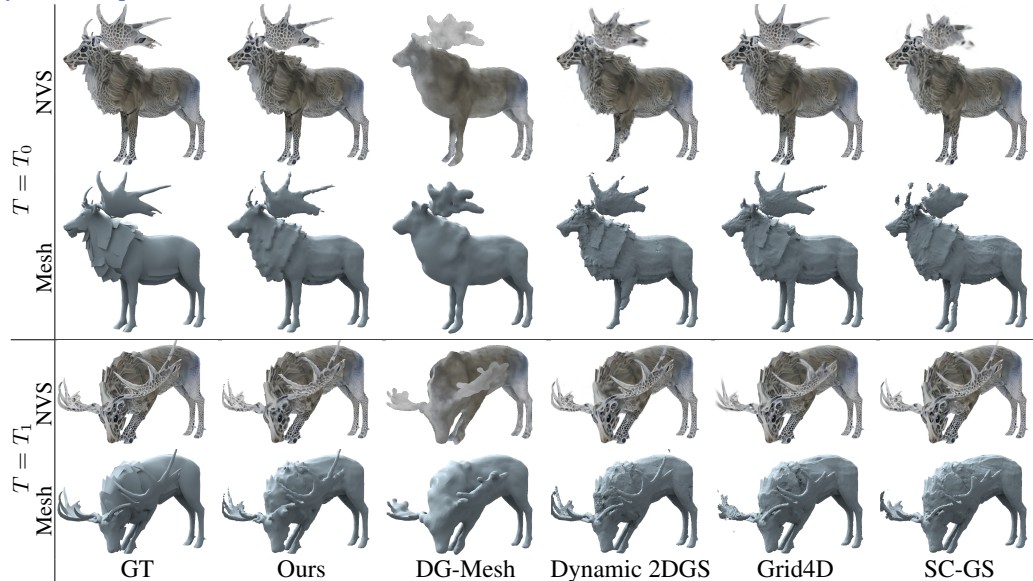

Figure 5: **Qualitative comparison on *SynthoMotion360* dataset.** We compare against baselines on both visual (visualized as shaded meshes) and geometry quality. Our method reconstructs realistic appearance and accurate geometry with fewer artifacts for large motions.

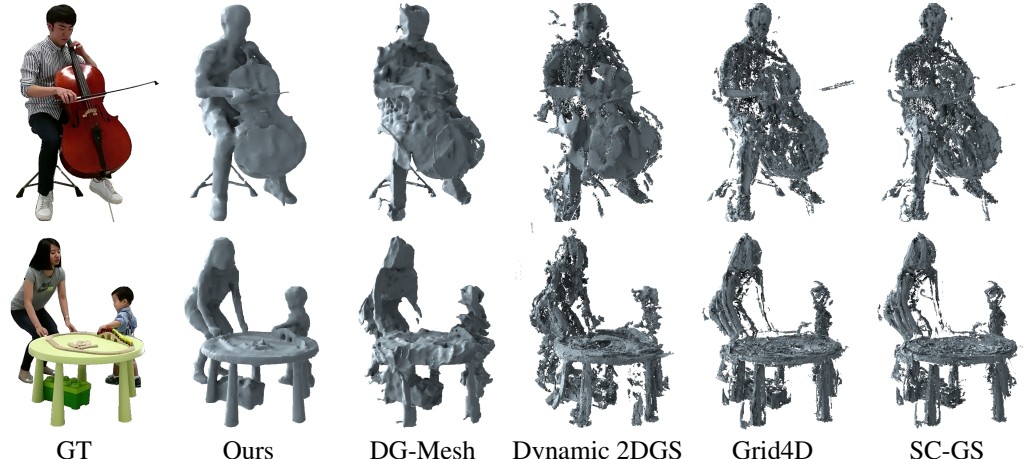

Figure 6: **Qualitative comparison on *CMU Panoptic Studio* dataset.** We compare the geometry quality against all baselines. Our method reconstructs accurate geometry, while baselines for surface reconstruction produce noticeable artifacts and broken geometry.

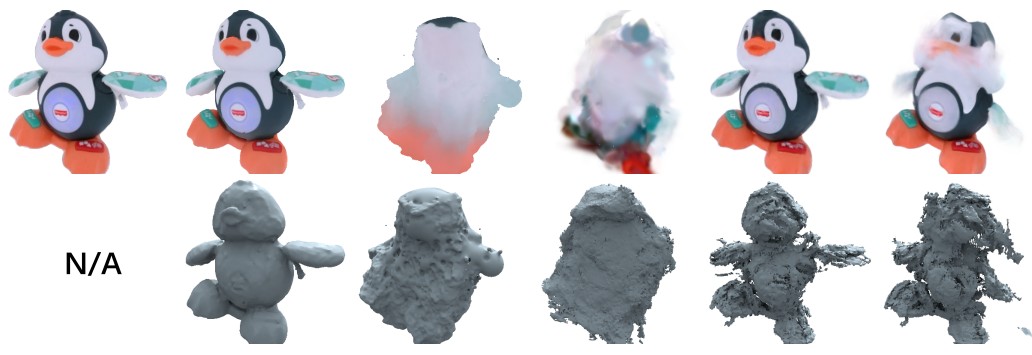

GT      Ours      DG-Mesh      Dynamic 2DGS      Grid4D      SC-GS

Figure 7: **Qualitative comparison on *DiVa-360* dataset.** Similarly, our method reconstructs accurate geometry, while baselines for surface reconstruction and dynamic NVS all produce noticeable artifacts and broken geometry.

## 4.3 GEOMETRY QUALITY

**Geometry Quality on *SynthoMotion360*.** On *SynthoMotion360* dataset that provides ground truth meshes, we can directly assess geometric fidelity by reporting Chamfer Distance (CD), F1 scores (F1), Edge Chamfer Distance (ECD), Edge F1 scores (EF1), and mesh statistics (#V, #F), as shown in Table 2. Our method consistently achieves state-of-the-art mesh quality. While per-frame approaches such as NeuS2 (Wang et al., 2023) yield competitive geometric metrics, they fundamentally lack temporal motion modeling with motion interpolation across contiguous timestamps. Other baselines originally designed for monocular settings still struggle to produce artifact-free geometry even when trained with multi-view inputs. As shown in F, while multi-view supervision improves their quantitative metrics (e.g., PSNR), it fails to resolve topological inconsistencies and high-frequency noise inherent to their representations. This performance gap highlights the inability of adapted monocular priors to handle complex topological changes and validates the necessity of a dedicated multi-view formulation for robust dynamic reconstruction in real-world scenarios.

**Geometry Quality on *CMU Panoptic Studio*.** Unlike *SynthoMotion360* dataset where the ground truth mesh is available for a comprehensive evaluation, *CMU Panoptic Studio* only provide scanned point cloud as geometry ground truth. We only assess geometry quality by Chamfer Distance and F1 scores. Additionally, we follow Shen et al. (2023), employing triangle quality metrics, such as triangle aspect ratios, radius ratios, and min angles to evaluate mesh quality. Tbl. 3 confirms the superior geometry quality of our method, as also evidenced by Fig. 6.

Table 2: **Geometry quality evaluation on *SynthoMotion-360* dataset** (*methods that are originally designed for monocular settings have been adapted to multi-view for fair comparisons).

| Method | Dynamics Modeling | CD ($10^{-3}$) ↓ | F1 ↑ | ECD ($10^{-2}$) ↓ | EF1 ↑ | #V | #F |
|---|---|---|---|---|---|---|---|
| Deformable-3DGS* | | 4.535 | 0.663 | 5.038 | 0.4312 | 107902 | 214957 |
| SC-GS* | | 3.051 | 0.770 | 3.898 | 0.5110 | 108999 | 217187 |
| Dynamic-2DGS* | Yes (Animatable) | 2.850 | 0.754 | 3.802 | 0.4939 | 124420 | 246261 |
| Grid4D* | | 4.637 | 0.700 | 4.708 | 0.4604 | 114096 | 226514 |
| DG-Mesh* | | 3.760 | 0.661 | 4.740 | 0.4382 | 42736 | 85447 |
| NeuS2 | No (Per-frame) | 2.253 | 0.908 | 3.220 | 0.5928 | 11820 | 23503 |
| AT-GS | | 2.339 | 0.865 | 3.227 | 0.5662 | 360767 | 721672 |
| Ours | Yes (Animatable) | 2.220 | 0.889 | 3.170 | 0.5955 | 14523 | 29041 |

Table 3: **Geometry quality evaluation on *CMU Panoptic Studio* dataset** (*methods that are originally designed for monocular settings have been adapted to multi-view for fair comparisons).

| Method | CD ($10^{-3}$) ↓ | F1 ↑ | Aspect> 4 (%) ↓ | Radius> 4 (%) ↓ | MinAngle< $10°$ (%) ↓ | #V | #F |
|---|---|---|---|---|---|---|---|
| Deformable-3DGS* | 2.749 | 0.887 | 22.00 | 26.59 | 12.00 | 667522 | 1247508 |
| SC-GS* | 2.372 | 0.893 | 21.93 | 26.51 | 11.97 | 655006 | 1230404 |
| Dynamic-2DGS* | 199.6 | 0.763 | 22.66 | 27.40 | 12.44 | 894010 | 1677482 |
| Grid4D* | 2.554 | 0.887 | 21.97 | 26.55 | 11.98 | 674237 | 1261074 |
| DG-Mesh* | 103.7 | 0.789 | 22.51 | 26.50 | 12.90 | 131361 | 262731 |
| Ours | 2.375 | 0.907 | 20.51 | 24.40 | 11.70 | 12802 | 25388 |

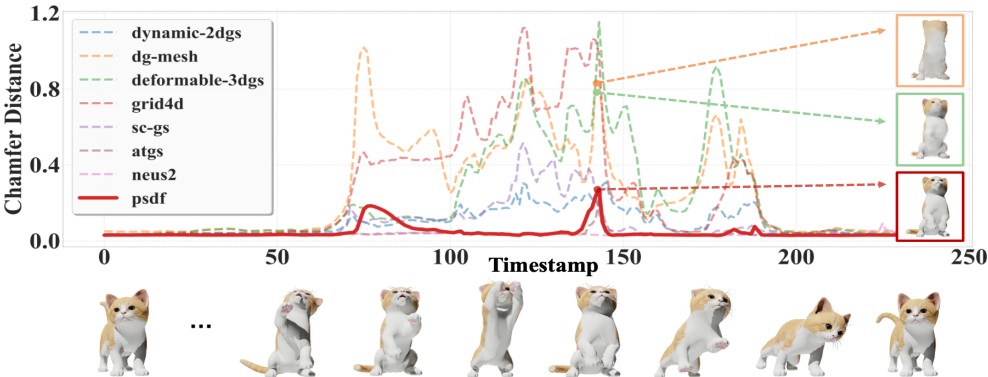

Figure 8: Temporal coherence evaluation on *SynthoMotion360*.

**Temporal Coherence on *SynthoMotion360*.** We further evaluate the temporal coherence of these dynamic mesh reconstruction approaches by assessing their chamfer distance variance across timestamps, as shown in Tbl. 4. Fig. 8 provides an illustration on the *Cat* case, demonstrating the superior ability of our parametric SDF representation in tackling complicated dynamics like large motions.

Table 4: Temporal coherence evaluation on *SynthoMotion360*.

| Method | CD Mean ↓ | CD Std. ↓ |
|---|---|---|
| Deformable-3DGS* | 0.0023 | 0.0025 |
| SC-GS* | 0.0009 | 0.0010 |
| Dynamic-2DGS* | 0.0009 | 0.0007 |
| Grid4D* | 0.0026 | 0.0030 |
| DG-Mesh* | 0.0026 | 0.0025 |
| Ours | 0.0005 | 0.0004 |

### 4.4 RELIGHTING

To further validate the accuracy and reliability of our dynamic reconstruction, we conduct relighting experiments using our reconstructed geometry and PBR material. As shown in Fig 9, the reconstructed results enable high-quality relighting for dynamic scenes.

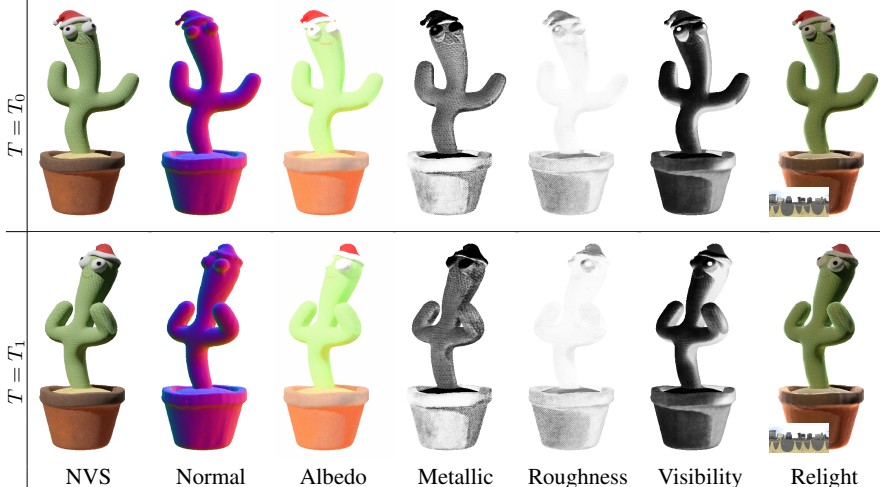

NVS    Normal    Albedo    Metallic    Roughness    Visibility    Relight

Figure 9: **PBR material reconstruction and relighting results.** Our method reconstructs plausible PBR materials (albedo, metallic and roughness) from multi-view videos, supporting re-lighting applications with new environment maps.

## 5 CONCLUSIONS AND LIMITATIONS

**Conclusions.** In this paper, we introduced parametric Signed Distance Function (p-SDF), a novel dynamic scene representation that leverages parametric basis functions to represent SDF trajectories. p-SDF naturally enforces temporally coherent geometry while remaining flexible to large deformations and topological changes. Combined with a physically based appearance modeling, our framework yields accurate geometry, robust material decomposition in highly dynamic conditions, and effectively supports downstream application such as relighting. Experiments on both synthetic and real-world datasets demonstrate substantial improvements over prior methods in geometric fidelity and visual quality. We believe p-SDF offers a promising direction for dynamic surface reconstruction and its applications.

**Limitations.** Our approach relies on multi-view video as input, which may limit its applicability in scenarios where only monocular data is available. In addition, the parametric SDF we used leverages the basis functions at each fixed spatial location, and thus, it does not explicitly represent the correspondence information along the time.

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

## APPENDIX

In this appendix, we provide supplementary material to support our main paper. We begin with a deeper analysis of our p-SDF representation in Sec. A by modeling a challenging topological changes, followed by the detailed mathematical formulations of the loss functions used for optimization in Sec. B. To further validate our method, we present extensive additional results in Sec. C and comprehensive ablation studies in Sec. D that justify our key design choices. Finally, we provide a detailed visualization of our new *SynthoMotion-360* benchmark in Sec. I, which was used for our evaluations.

## A  MODELING TOPOLOGICAL CHANGES WITH P-SDF

A key challenge in dynamic surface reconstruction is handling topological changes, such as objects merging or holes closing. In this section, we demonstrate how our parametric SDF (p-SDF) representation is capable of modeling these abrupt events.

We posit that a topological change at a specific location in space manifests as a near-instantaneous discontinuity in the temporal SDF trajectory of that point. To analyze this, we use the canonical example of a **torus closing its hole to become a sphere**. For any point originally inside the hole, its SDF value—representing the shortest distance to the surface—must abruptly transition from a positive value (outside the object's mass) to a negative value (inside the object's mass) at the exact moment the hole closes.

First, we validate the expressiveness of our p-SDF representation itself. *Can our basis function formulation capture such a sharp discontinuity?* To test this, we generated a ground-truth (GT) SDF temporal trajectory for a shape undergoing this torus-to-sphere transition. In Fig. A1, the GT SDF curve (red) shows a sharp, step-like drop, which corresponds to the moment of topological change. We then optimize the parameters of our p-SDF curve representation to fit this ground-truth signal. The resulting predicted SDF curve (blue) closely matches the GT, successfully capturing the sharp discontinuity. This demonstrates that our parametric representation is flexible enough to model the drastic SDF variations that define topological changes.

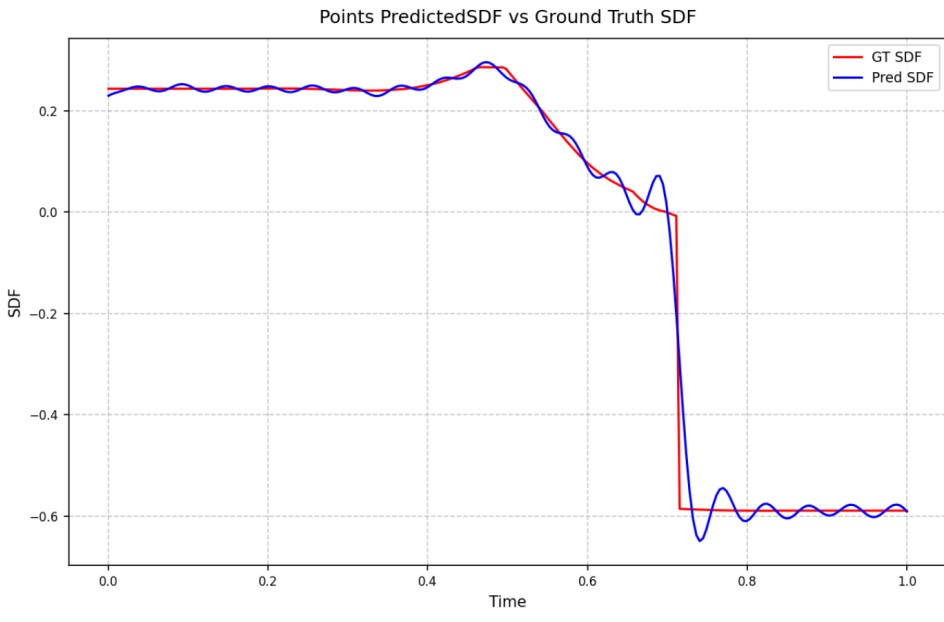

Figure A1: **Fitting an Topological Change.** An analysis of SDF values for a point in space as a torus's hole closes. The Ground Truth SDF (red) exhibits a sharp drop, representing the topological change. Our p-SDF representation (blue), when optimized to fit this signal, accurately captures this sharp transition, demonstrating its expressive power to model such events.

Beyond the representation's capacity, we further demonstrate that our full framework can reconstruct such events when trained only with indirect image supervision. We created a synthetic "torus-to-sphere" sequence and applied our full optimization pipeline. As illustrated in Fig. A2, our method successfully reconstructs the entire dynamic process.

This validation confirms that our p-SDF framework is not only theoretically capable of representing topological changes but can also be effectively optimized to reconstruct them in practice.

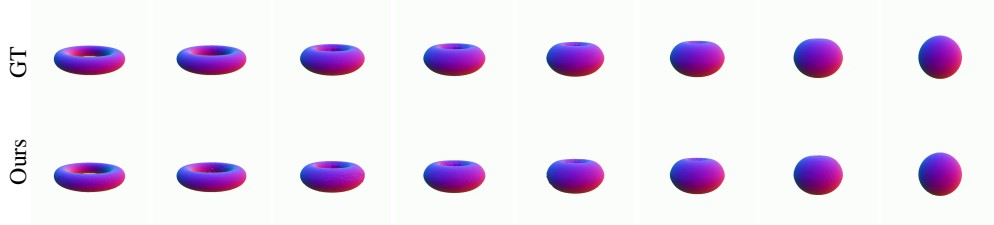

Figure A2: **Reconstruction of a Topological Change.** Our method, trained with only image-based supervision, successfully reconstructs the "torus-to-sphere" transition. The sequence shows the hole smoothly closing over time, demonstrating the practical effectiveness of our approach in handling topological changes.

## B    DETAILS OF LOSS FUNCTIONS

In this section, we provide the detailed formulation for the loss functions summarized in the main paper. Our total training objective is a weighted sum of three main components: a photometric loss $\mathcal{L}_{\text{photo}}$, a geometric regularization loss $\mathcal{L}_{\text{geo}}$, and an appearance regularization loss $\mathcal{L}_{\text{app}}$.

### B.1    PHOTOMETRIC LOSS

During training, given a viewpoint $v$, its corresponding ground-truth image $I_v$ and mask $A_v$ we differentiably render an image $\hat{I}_v$ and mask $\hat{A}_v$, and the photometric loss is defined as:

$$
\begin{aligned}
\mathcal{L}_{\text{photo}} &= \lambda_{\text{L1}}\mathcal{L}_{\text{L1}} + \lambda_{\text{ssim}}\mathcal{L}_{\text{SSIM}} + \lambda_{\text{mask}}\mathcal{L}_{\text{mask}} \\
&= \|\hat{I}_v - I_v\|_1 + \left(1 - \text{SSIM}(\hat{I}_v, I_v)\right) + (\hat{A}_v - A_v)^2.
\end{aligned}
\tag{A1}
$$

In experiments, we set $\lambda_{\text{L1}}$, $\lambda_{\text{ssim}}$, and $\lambda_{\text{mask}}$ to 0.1, 0.1, 0.3, respectively.

### B.2    GEOMETRIC REGULARIZATION LOSS

To guide the optimization towards physically plausible surfaces, we introduce a composite geometric regularization terms, $\mathcal{L}_{\text{geo}}$, which is defined as:

$$
\mathcal{L}_{\text{geo}} = \lambda_{\text{eik}}\mathcal{L}_{\text{eik}} + \lambda_{\text{deriv}}\mathcal{L}_{\text{deriv}} + \lambda_{\text{sdf}}\mathcal{L}_{\text{sdf}} + \lambda_{\text{lap}}\mathcal{L}_{\text{lap}}.
\tag{A2}
$$

In this equation, $\mathcal{L}_{\text{eik}}$ denotes the Eikonal loss, which ensures the learned function $s(\mathbf{x}, t)$ is a valid signed distance function by encouraging its spatial gradient norm to be a unit. Specifically, for a set of points $\mathcal{X}$ randomly sampled in the 3D space at an arbitrary time $t$, the Eikonal loss is defined as:

$$
\mathcal{L}_{\text{eik}} = \frac{1}{|\mathcal{X}|} \sum_{\mathbf{x} \in \mathcal{X}, t} \left(\|\nabla_{\mathbf{x}} s(\mathbf{x}, t)\|_2 - 1\right)^2.
\tag{A3}
$$

$\mathcal{L}_{\text{deriv}}$ penalizes the magnitude of the SDF's first-order temporal derivative at each grid vertex and encourages smooth motion between adjacent grid vertices $(v_i, v_j)$. It's defined as:

$$
\mathcal{L}_{\text{deriv}} = \frac{1}{|V_{\text{grid}}|} \sum_{v_i} \int_{t \in T} \left\|\frac{\partial s(v_i, t)}{\partial t}\right\|_2^2 dt + \sum_{(v_i, v_j)} \int_{t \in T} \left\|\frac{\partial s(v_i, t)}{\partial t} - \frac{\partial s(v_j, t)}{\partial t}\right\|_2^2 dt.
\tag{A4}
$$

We further add SDF surface regularization term $\mathcal{L}_{\text{sdf}}$ and Laplacian smoothness loss $\mathcal{L}_{\text{lap}}$ to regularize the smoothness of mesh surface at every timestep, following the definition from Flexi-Cubes (Shen et al., 2023).

Note that, the regularization terms we add are mainly to enforce the spatial smoothness of the mesh geometry, the temporal smoothness is implicitly regularized by our parametric SDF representation.

In experiments, we set $\lambda_{\text{eik}}$, $\lambda_{\text{deriv}}$, $\lambda_{\text{sdf}}$, and $\lambda_{\text{lap}}$ to 0.01, 0.05, 0.05, 0.01, respectively.

### B.3 APPEARANCE REGULARIZATION LOSS

Following Nvdiffrec (Munkberg et al., 2022), we apply appearance regularization term $\mathcal{L}_{\text{app}}$ on albedo, roughness and metallic. Specifically, it is defined as:

$$\mathcal{L}_{\text{app}} = \lambda_{\text{feat}}\mathcal{L}_{\text{feat}} + \lambda_{\text{light}}\mathcal{L}_{\text{light}}. \tag{A5}$$

Here, $\mathcal{L}_{\text{feat}}$ prevents noisy or aliased textures by applying a spatial smoothness prior to the features within the three 3D hash grids ($G_{xyt}$, $G_{xzt}$, $G_{yzt}$). The loss is defined as:

$$\begin{aligned}
\mathcal{L}_{\text{feat}} =& \mathbb{E}_{\mathbf{x},\delta}\left[\|G_{xyt}(\mathbf{x}) - G_{xyt}(\mathbf{x}+\delta)\|_2^2\right] + \\
& \mathbb{E}_{\mathbf{x},\delta}\left[\|G_{xzt}(\mathbf{x}) - G_{xzt}(\mathbf{x}+\delta)\|_2^2\right] + \\
& \mathbb{E}_{\mathbf{x},\delta}\left[\|G_{yzt}(\mathbf{x}) - G_{yzt}(\mathbf{x}+\delta)\|_2^2\right]
\end{aligned} \tag{A6}$$

where $\delta$ is a small random spatial offset.

$\mathcal{L}_{\text{light}}$ follows the white-balance strategy from Nvdiffrec and we refer the readers to the original paper for the defination. In experiments, we set $\lambda_{\text{feat}}$ and $\lambda_{\text{light}}$ to 0.001, 0.01, respectively.

## C ADDITIONAL

In this section, we show additional detailed results of the evaluation in our main paper.

### C.1 RESULTS ON SYNTHOMOTION360 DATASET

**Quantitative Results**    Tab. A1 presents the per-scene quantitative comparison on this dataset. Our method consistently outperforms prior works across most test scenes.

**Qualitative Results**    Fig. A3 and A4 provides per-scene qualitative comparisons, where we show both the rendered novel views and the extracted surface meshes. Noticeably, p-SDF excels at capturing fine geometric details and maintaining temporal consistency where others fail.

### C.2 RESULTS ON CMU PANOPTIC 360 DATASET

**Qualitative Results**    Fig. A5 provides per-scene qualitative comparisons, where we show the extracted surface meshes. p-SDF can faithfully reconstruct smooth results with dynamic details, which are often distorted in other methods.

### C.3 RESULTS ON DIVA-360 DATASET

**Quantitative Results**    Tab. A2 presents the per-scene quantitative comparison on this dataset. Our method achieves comparable performance with state-of-the-art works across all test scenes.

**Qualitative Results**    Fig. A6 provides per-scene qualitative comparisons, where we show both the rendered novel views and the extracted surface meshes. Though some methods succeed in synthesizing novel views, p-SDF successfully reconstructs delicate geometry while all other methods fail.

Table A1: Quantitative evaluation of our method compared to previous work on every scene of *SynthoMotion-360* datasets. For clarity, all Chamfer distances are scaled by $10^{-3}$.

| Scene | Method | PSNR↑ | SSIM↑ | LPIPS↓ | MAE↓ | Chamfer↓ |
|---|---|---|---|---|---|---|
| Cat | Deformable-3DGS | 30.425 | 0.970 | 0.071 | 2.072 | 2.272 |
| | SC-GS | 31.654 | 0.972 | 0.058 | 1.726 | 0.923 |
| | Dynamic-2DGS | 31.098 | 0.970 | 0.062 | 1.988 | 0.940 |
| | Grid4D | 31.027 | 0.971 | 0.065 | 2.140 | 2.557 |
| | DG-Mesh | 26.882 | 0.956 | 0.068 | 2.916 | 2.608 |
| | **Ours** | **34.407** | **0.984** | **0.024** | **0.751** | **0.456** |
| Deer | Deformable-3DGS | 27.035 | 0.935 | 0.096 | 4.209 | 3.184 |
| | SC-GS | 29.298 | 0.954 | 0.055 | 3.651 | 0.639 |
| | Dynamic-2DGS | 28.936 | 0.950 | 0.060 | 3.344 | 0.368 |
| | Grid4D | **30.340** | **0.962** | 0.047 | 3.280 | 0.342 |
| | DG-Mesh | 24.041 | 0.882 | 0.149 | 3.495 | 1.379 |
| | **Ours** | 29.128 | 0.956 | **0.039** | **2.518** | **0.204** |
| Football Player | Deformable-3DGS | 29.912 | 0.975 | 0.049 | 1.440 | 0.492 |
| | SC-GS | 29.750 | 0.977 | 0.041 | 1.301 | 0.571 |
| | Dynamic-2DGS | 30.702 | 0.978 | 0.039 | 1.281 | 0.232 |
| | Grid4D | 30.949 | 0.980 | 0.036 | 1.221 | 0.226 |
| | DG-Mesh | 26.950 | 0.963 | 0.043 | 1.131 | 0.243 |
| | **Ours** | **31.275** | **0.982** | **0.019** | **0.877** | **0.194** |
| Lego | Deformable-3DGS | 23.542 | 0.933 | 0.118 | 5.495 | 23.261 |
| | SC-GS | 24.459 | 0.939 | 0.095 | 4.548 | 17.703 |
| | Dynamic-2DGS | 23.421 | 0.929 | 0.106 | 5.078 | 17.173 |
| | Grid4D | 23.636 | 0.934 | 0.113 | 5.562 | 25.859 |
| | DG-Mesh | 19.297 | 0.906 | 0.132 | 6.851 | 20.645 |
| | **Ours** | **27.032** | **0.957** | **0.043** | **2.754** | **14.279** |
| Rabbit | Deformable-3DGS | 30.492 | 0.955 | 0.082 | 2.521 | 0.546 |
| | SC-GS | 31.895 | 0.963 | 0.067 | 2.009 | 0.344 |
| | Dynamic-2DGS | 30.457 | 0.956 | 0.077 | 2.528 | 0.688 |
| | Grid4D | 30.961 | 0.958 | 0.074 | 2.496 | 0.709 |
| | DG-Mesh | 27.534 | 0.945 | 0.073 | 2.360 | 0.769 |
| | **Ours** | **33.070** | **0.975** | **0.038** | **1.051** | **0.297** |
| Spiderman Fight | Deformable-3DGS | 28.221 | 0.960 | 0.068 | 2.043 | 0.526 |
| | SC-GS | 29.037 | 0.967 | 0.049 | 1.576 | 0.186 |
| | Dynamic-2DGS | 28.533 | 0.963 | 0.051 | 1.922 | 0.428 |
| | Grid4D | 30.235 | 0.970 | 0.049 | 1.638 | 0.137 |
| | DG-Mesh | 24.650 | 0.946 | 0.051 | 2.029 | 0.528 |
| | **Ours** | **30.321** | **0.982** | **0.016** | **0.910** | **0.034** |
| Toy | Deformable-3DGS | 32.650 | 0.962 | 0.051 | 2.112 | 1.466 |
| | SC-GS | 33.073 | 0.962 | 0.040 | 1.794 | 0.992 |
| | Dynamic-2DGS | 33.275 | 0.965 | 0.037 | 1.089 | 0.120 |
| | Grid4D | 33.184 | 0.962 | 0.041 | 2.137 | 2.629 |
| | DG-Mesh | 30.678 | 0.956 | 0.048 | 0.778 | 0.150 |
| | **Ours** | **36.337** | **0.980** | **0.011** | **0.465** | **0.084** |

# D    ABLATION STUDY

In this section, we conduct a series of ablation studies to analyze the impact of our key design choices. We aim to validate the effectiveness of: (1) our mixed basis function for modeling geometry, (2) our factorized representation for the dynamic appearance field, and (3) our use of temporal regularization for ensuring motion smoothness. Unless otherwise specified, all ablation experiments are conducted on the "Toy" scene from the *SynthoMotion-360* dataset.

## D.1    EFFECT OF BASIS FUNCTION COMPOSITION

Our parametric SDF models the temporal evolution of geometry using a set of basis functions $B(t)$. We propose a mixed basis that combines low-degree polynomials with Fourier features to capture both low-frequency, coarse motions and high-frequency, detailed dynamics. To validate this design, we compare our full model against two variants:

- **Polynomial Only.** Where $B(t)$ only consists of polynomial basis functions.

- **Fourier Only.** Where $B(t)$ consists exclusively of Fourier frequency basis functions.

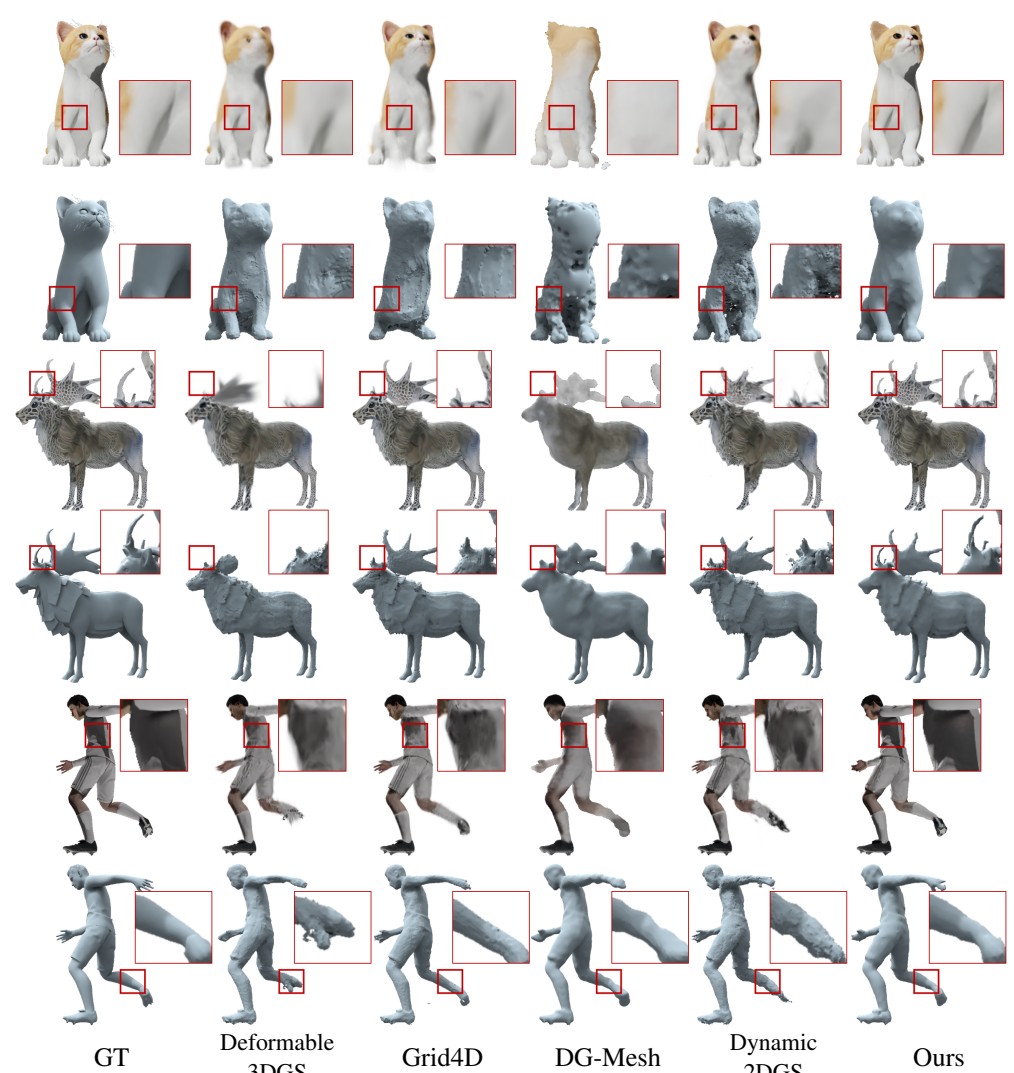

Figure A3: Qualitative results on SynMotion360 (Part I). We compare against baselines on both novel view synthesis and geometry reconstruction (visualized as shaded meshes). Our method reconstructs smoother and more accurate models with fewer artifacts.

As shown in Tab. A3, the choice of basis is critical. The Polynomial Only variant suffers a catastrophic drop in performance across all metrics; for instance, the PSNR decreases by over 10 dB and the Chamfer distance increases by nearly 30-fold. This confirms that low-degree polynomials alone are entirely insufficient to capture the high-frequency dynamics of the scene, leading to overly smoothed and geometrically incorrect reconstructions. Conversely, the Fourier Only model performs very strongly, achieving results close to our full model. This indicates that Fourier features are the primary driver for modeling complex dynamics. Nonetheless, our mixed basis still yields a consistent improvement across all metrics, suggesting that the polynomial component effectively models the low-frequency motion backbone, allowing the Fourier features to more accurately refine the high-frequency details.

## D.2 EFFECT OF APPEARANCE REGULARIZATION

To prevent noisy textures and encourage a smooth appearance field, we apply a smoothness regularization to the features stored in our dynamic appearance grid. We analyze its impact by training a model without this component (**w/o Hash Grid Reg.**).

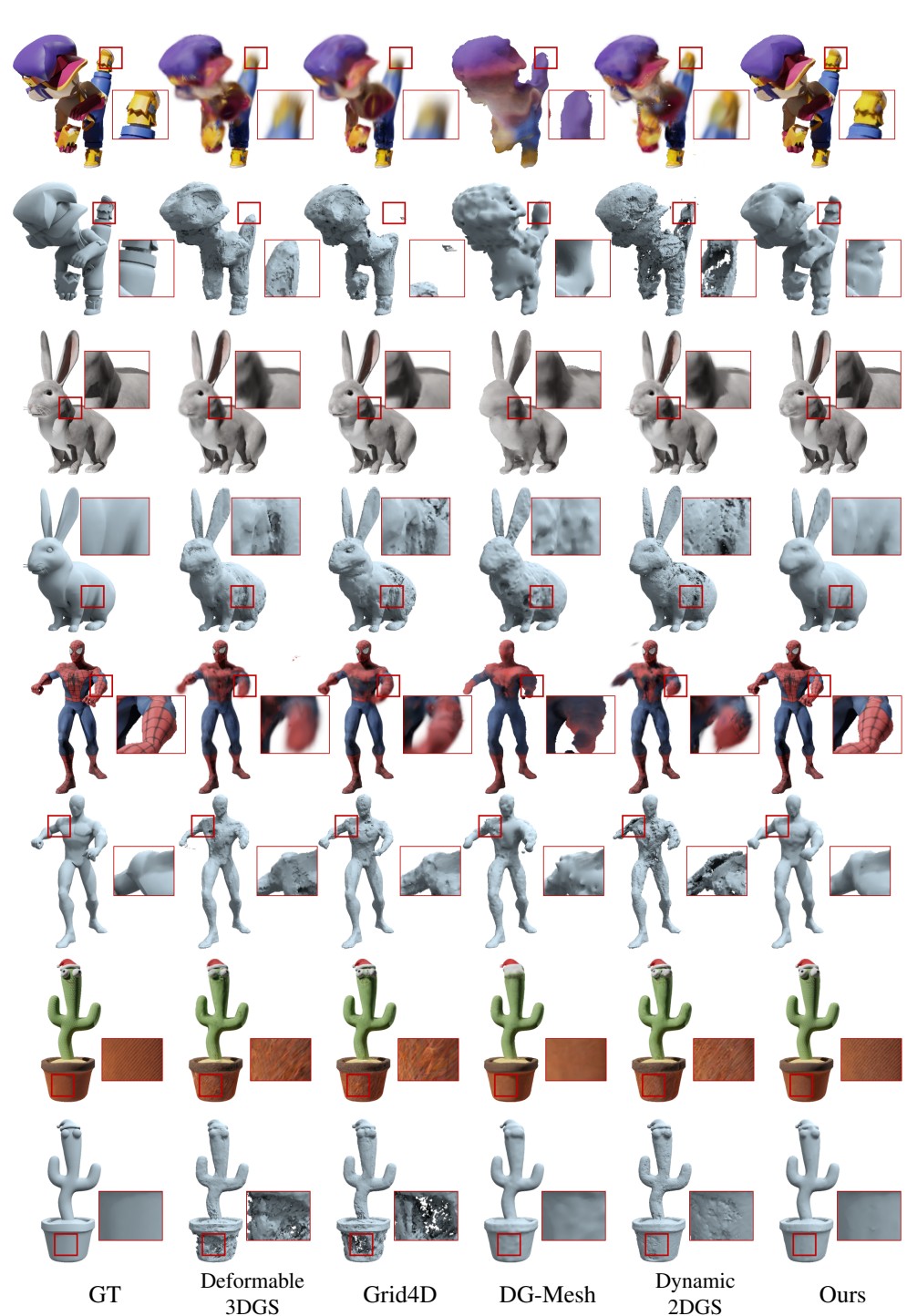

Figure A4: Qualitative results on SynMotion360 (Part II). We compare against baselines on both novel view synthesis and geometry reconstruction (visualized as shaded meshes). Our method reconstructs smoother and more accurate models with fewer artifacts.

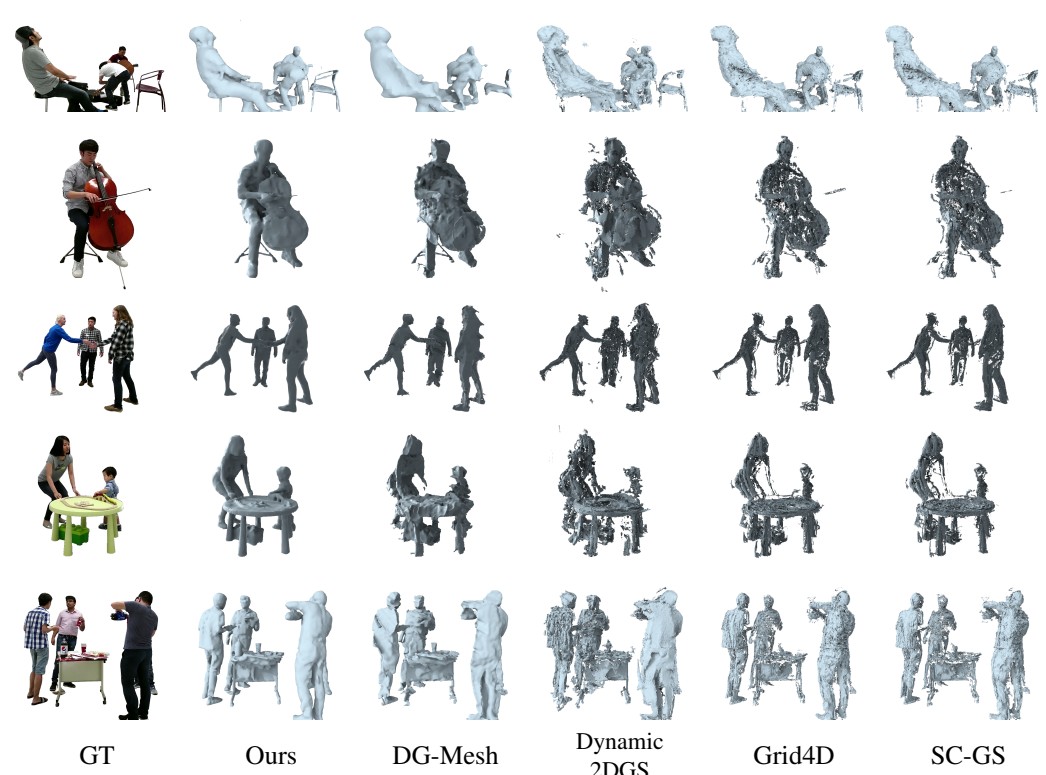

| GT | Ours | DG-Mesh | Dynamic 2DGS | Grid4D | SC-GS |

Figure A5: Qualitative results on CMU360. Our method successfully captures the complex, non-rigid human motion and produces temporally coherent surface reconstructions.

The quantitative results in Tab. A3 show only a minor decrease in performance. This is because the primary role of this regularization is to improve qualitative results rather than single-frame metrics. Without it, the learned materials can exhibit slight noise and aliasing artifacts, particularly on textured surfaces. This simple prior proves beneficial for obtaining clean, plausible materials and robustly disentangling them from scene lighting.

### D.3 IMPACT OF TEMPORAL REGULARIZATION

To ensure the learned motion is smooth and physically plausible, we introduce a temporal regularization loss that promotes coherent dynamics. We ablate this component by training our model without this loss term (**w/o Temporal Reg.**).

As the quantitative results in Tab. A3 indicate, removing this regularization leads to only a marginal drop in per-frame metrics. This is expected, as these metrics do not evaluate temporal consistency. Qualitatively, however, the impact is significant. Without this term, the reconstructed surface exhibits noticeable jitter and flickering, especially during fast motions, as the mesh vertices no longer follow smooth paths. This demonstrates that the temporal regularization is essential for producing the smooth, coherent dynamics that are critical for high-quality animation and downstream applications.

### D.4 ABLATION STUDY ON GRID RESOLUTION

To analyze the impact of spatial discretization on reconstruction quality, we conducted an ablation study on the grid resolution of our p-SDF representation. We evaluated three resolution settings: $64^3$, $96^3$ (our default), and $128^3$, using the *Deer* scene from the *SynthoMotion-360* dataset.

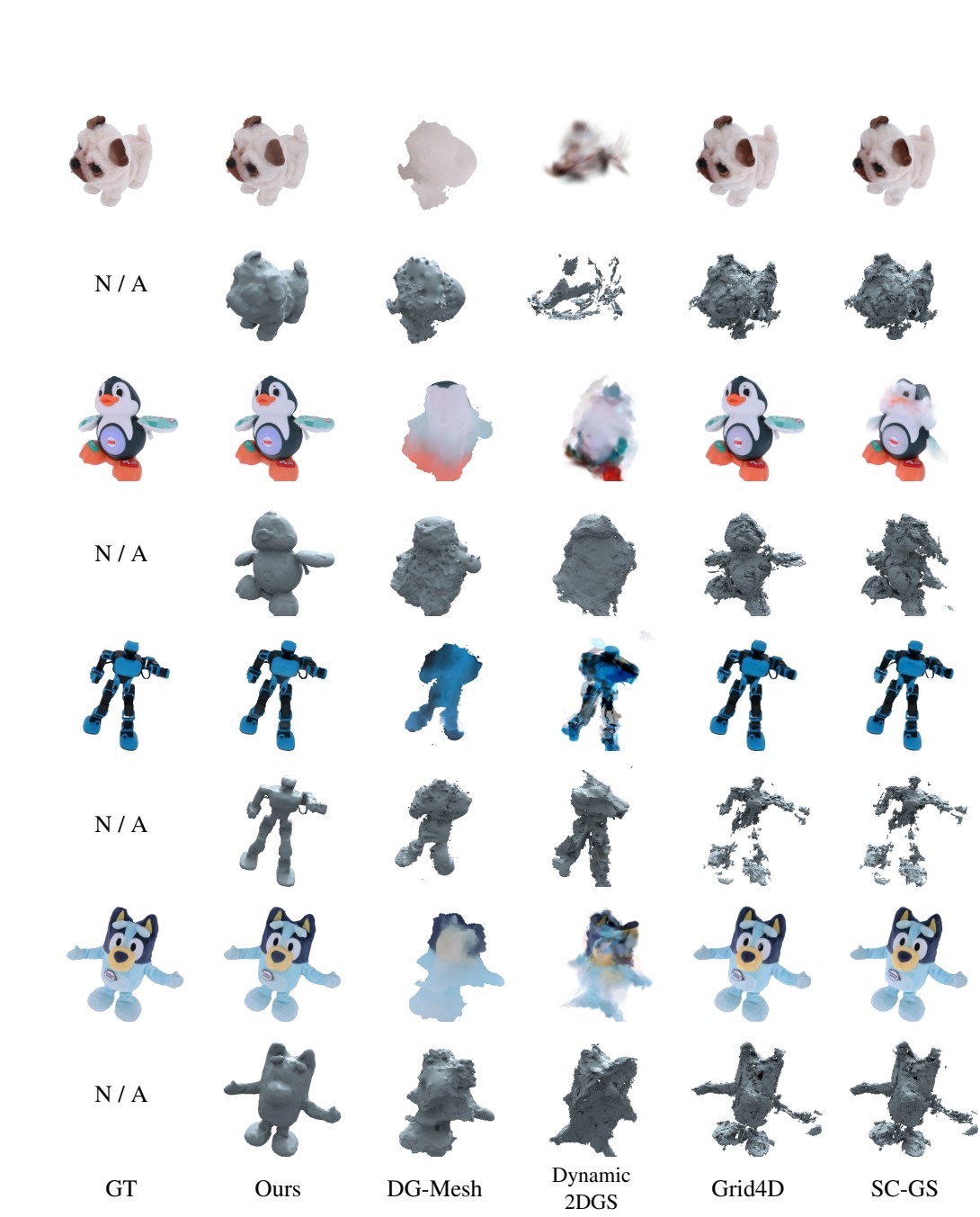

GT    Ours    DG-Mesh    Dynamic 2DGS    Grid4D    SC-GS

Figure A6: Qualitative results on DIVA-360. Our method reconstructs significantly smoother and delicate geometry.

Table A2: Quantitative evaluation of our method compared to prior works on every scene of *DiVa-360* datasets.

| Scene | Method | PSNR↑ | SSIM↑ | LPIPS↓ |
|---|---|---|---|---|
| Dog | Deformable-3DGS | 28.556 | **0.964** | 0.098 |
| | SC-GS | **29.181** | **0.964** | 0.084 |
| | Dynamic-2DGS | 20.868 | 0.947 | 0.164 |
| | Grid4D | 28.362 | 0.961 | 0.099 |
| | DG-Mesh | 21.362 | 0.944 | 0.162 |
| | Ours | 29.132 | **0.964** | **0.058** |
| K1 Double Punch | Deformable-3DGS | 24.751 | **0.946** | 0.051 |
| | SC-GS | 24.383 | 0.944 | **0.048** |
| | Dynamic-2DGS | 13.710 | 0.864 | 0.197 |
| | Grid4D | 24.615 | 0.943 | 0.049 |
| | DG-Mesh | 14.155 | 0.889 | 0.183 |
| | Ours | **27.135** | 0.944 | **0.048** |
| Penguin | Deformable-3DGS | 28.557 | **0.963** | 0.050 |
| | SC-GS | 21.472 | 0.943 | 0.104 |
| | Dynamic-2DGS | 17.157 | 0.912 | 0.187 |
| | Grid4D | **28.611** | 0.960 | 0.048 |
| | DG-Mesh | 16.986 | 0.919 | 0.185 |
| | Ours | 27.848 | 0.962 | **0.040** |
| Wolf | Deformable-3DGS | 27.255 | 0.953 | 0.072 |
| | SC-GS | 27.857 | 0.953 | 0.064 |
| | Dynamic-2DGS | 18.320 | 0.916 | 0.198 |
| | Grid4D | 28.044 | 0.952 | 0.065 |
| | DG-Mesh | 18.435 | 0.924 | 0.203 |
| | Ours | **28.189** | **0.957** | **0.051** |

Table A3: Quantitative results of our ablation studies. We report key metrics for novel view synthesis and geometric reconstruction. Our full model consistently performs the best.

| Method | PSNR↑ | SSIM↑ | LPIPS↓ | MAE↓ | Chamfer↓ |
|---|---|---|---|---|---|
| (a) Polynomial Only | 25.910 | 0.952 | 0.049 | 2.492 | 2.438 |
| (b) Fourier Only | 35.904 | 0.978 | 0.012 | 0.505 | 0.086 |
| (c) w/o Temporal Reg. | 35.960 | 0.978 | 0.012 | 0.519 | 0.089 |
| (c) w/o Hash Grid Reg. | 35.893 | 0.978 | 0.012 | 0.522 | 0.091 |
| Ours (Full Model) | **36.337** | **0.980** | **0.011** | **0.465** | **0.084** |

Table A4 summarizes the geometric accuracy and novel view synthesis quality across different resolutions. The results illustrate a clear trade-off between reconstruction fidelity and grid density:

**Significant Gain from Coarse to Medium** ($64 \rightarrow 96$): Increasing the resolution from 64 to 96 yields substantial improvements. Geometric error (Chamfer Distance) decreases by approximately **35%**, and rendering quality (PSNR) increases by **1.59 dB**. This indicates that a resolution of 64 is too coarse to adequately capture high-frequency surface details.

**Diminishing Returns at High Resolution** ($96 \rightarrow 128$): Further increasing the resolution to 128 provides marginal gains in geometry. The Chamfer distance reduction slows down (approx. **5%** improvement), and surface normal accuracy plateaus. While rendering quality continues to improve (PSNR +0.88 dB), this comes at the cost of cubic growth in memory consumption.

**Conclusion:** Based on these observations, we selected **96** as our default resolution. It offers the optimal balance, achieving high-fidelity geometry and appearance comparable to finer grids while maintaining manageable computational costs.

Table A4: **Impact of Grid Resolution (Deer Scene).** Increasing resolution significantly improves quality up to a point, after which geometric gains diminish.

| Resolution | Chamfer ($\downarrow$) | Normal MAE ($\downarrow$) | PSNR ($\uparrow$) | SSIM ($\uparrow$) | LPIPS ($\downarrow$) |
|---|---|---|---|---|---|
| **64** | $3.11 \times 10^{-4}$ | 2.62 | 25.43 | 0.944 | 0.051 |
| **96 (Default)** | $2.03 \times 10^{-4}$ | **2.54** | 27.02 | 0.956 | 0.039 |
| **128** | $\mathbf{1.93 \times 10^{-4}}$ | 2.55 | **27.90** | **0.962** | **0.034** |

# E  EVALUATION ON TEMPORAL INTERPOLATION CAPABILITY

To further validate the continuous nature of our Parametric SDF (p-SDF) representation, we conducted an additional experiment focusing on **temporal interpolation**. This experiment evaluates the model's ability to reconstruct geometry and synthesize novel views at timestamps that were *not observed* during training.

We selected two representative scenes from the SynthoMotion-360 dataset to analyze performance under different motion characteristics: **"Toy"** represents smooth and moderate motion. **"Spiderman"** represents large-scale non-rigid deformation.

**Protocol.** We adopt a temporal hold-out protocol. The model is trained using only **even** frames ($t = 0, 2, 4, \dots$) from the multi-view video sequences. Evaluation is performed on the held-out **odd** frames ($t = 1, 3, 5, \dots$), which act as unseen intermediate timestamps. We report the Normal MAE and Chamfer Distance (CD) for geometric accuracy and PSNR for novel view synthesis quality.

Table A5 summarizes the quantitative comparison between seen timestamps (interpolation source) and unseen timestamps (interpolation target). We also visualize the frame-wise PSNR trajectories in Figure A7.

Table A5: **Quantitative results of Temporal Interpolation.** We compare the reconstruction quality on trained (even) frames versus interpolated (odd) frames.

| Scene | Split | CD ($\downarrow$) | PSNR ($\uparrow$) | Normal MAE ($\downarrow$) |
|---|---|---|---|---|
| **Toy** | Trained Frames (Even) | $8.68 \times 10^{-5}$ | 36.07 | 0.52 |
| | **Interpolated Frames (Odd)** | $\mathbf{9.68 \times 10^{-5}}$ | **32.90** | **0.66** |
| **Spiderman** | Trained Frames (Even) | $3.78 \times 10^{-5}$ | 30.00 | 0.98 |
| | **Interpolated Frames (Odd)** | $6.55 \times 10^{-3}$ | 18.90 | 4.46 |

**1. Effective Interpolation for Smooth Motion:** In the "Toy" scene, our method demonstrates excellent temporal generalization. As shown in Figure A7 (Top), the geometric error on unseen frames ($9.68 \times 10^{-5}$) is comparable to that of seen frames ($8.68 \times 10^{-5}$), and visual quality remains high (PSNR $> 32$ dB). This confirms that our p-SDF successfully learns the underlying continuous temporal trajectory of the surface using the polynomial and Fourier basis functions.

**2. Limitations under Extreme Dynamics:** In the "Spiderman" scene, which features rapid and large-scale deformations, we observe a performance drop on unseen frames, as shown in Figure A7 (Bottom). This is an expected phenomenon caused by temporal aliasing; when the motion between frames is too large to be inferred from the boundary conditions, interpolation naturally degrades.

**Conclusion:** These findings validate that p-SDF serves as a strong temporal regularizer. While it enables robust interpolation for continuous dynamics, high-fidelity reconstruction of extremely fast motions still benefits from the dense temporal observations used in our main experimental protocol.

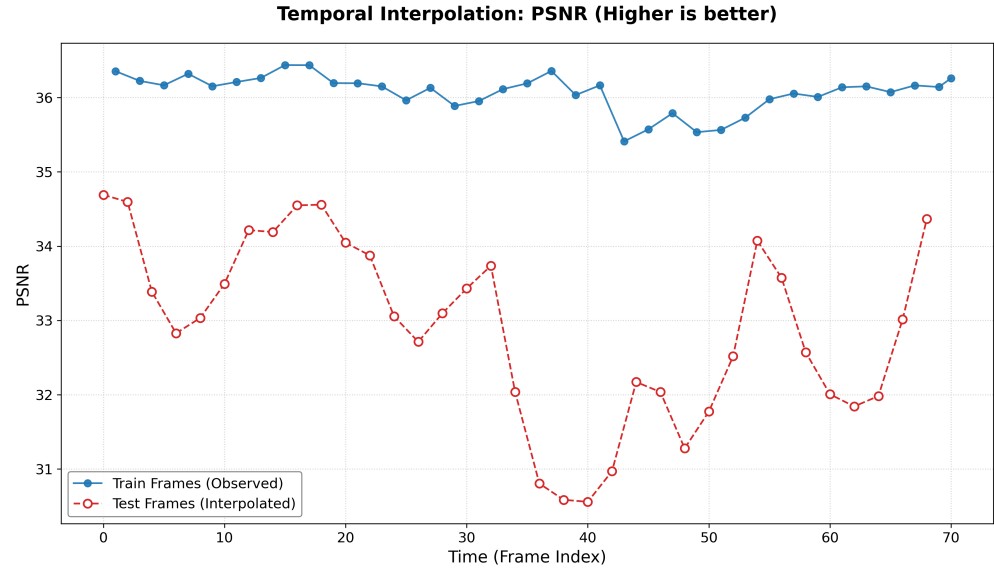

Figure A7: **Temporal Interpolation Analysis.** We plot the PSNR of training frames (Even, Blue) and interpolated test frames (Odd, Red) over time. **Top (Toy):** For smooth motion, p-SDF interpolates well, maintaining high PSNR. **Bottom (Spiderman):** For extreme motion, interpolation degrades, highlighting the need for sufficient temporal sampling.

## F  ANALYSIS OF MONOCULAR-BASED BASELINE PERFORMANCE UNDER MULTI-VIEW SUPERVISION

To explicitly address concerns regarding the fairness of comparing our method against baselines originally designed for monocular inputs (e.g., DG-Mesh, Grid4D), we conducted a controlled experiment to verify their performance behaviors under **Monocular** versus **Multi-view** settings.

We selected two representative baselines, **DG-Mesh** and **Grid4D**, and evaluated them on three scenes from the SynthoMotion-360 dataset (*Cat*, *Spiderman*, *Toy*) with varying motion complexities.

- **Monocular Setting:** The models were trained using a single camera view at each times-tamp to simulate their original design constraints.

- **Multi-view Setting (Used in Main Paper):** The models were retrained using the **exact same multi-view supervision** as our method, aggregating gradients from all available training cameras.

Table A6 details the performance metrics (PSNR, SSIM, LPIPS) for both settings. The results lead to the key conclusions: All baselines exhibit significant quantitative improvements (e.g., Grid4D gains +4.87 dB on Spiderman) when transitioning from monocular to multi-view supervision. This confirms that these representations are not inherently limited to monocular data. However, despite these metric gains, they still fail to resolve high-frequency geometric artifacts and temporal jitter (as seen in Fig. A4). The comparisons presented in our main paper utilize the Multi-view performance numbers (the stronger setting). By comparing our method against the "upper bound" performance of these baselines, we ensure the evaluation is strictly fair.

Table A6: **Impact of Multi-view Supervision on Baselines.** We compare the performance of baselines when trained with Monocular vs. Multi-view data. The results demonstrate that these methods naturally extend to multi-view settings and achieve significantly higher performance. **Note:** The results reported in our main paper correspond to the **"Multi-view"** (Best-case) rows.

| Method | Scene | Setting | PSNR ($\uparrow$) | SSIM ($\uparrow$) | LPIPS ($\downarrow$) | Improvement |
|--------|-------|---------|-------|-------|--------|-------------|
| DG-Mesh | Cat | Monocular | 24.06 | 0.9479 | 0.0817 | |
| | | **Multi-view** | **26.88** | **0.9559** | **0.0635** | **+2.82 dB** |
| | Spiderman | Monocular | 22.68 | 0.9359 | 0.0700 | |
| | | **Multi-view** | **24.66** | **0.9463** | **0.0589** | **+1.98 dB** |
| | Toy | Monocular | 29.00 | 0.9500 | 0.0524 | |
| | | **Multi-view** | **30.68** | **0.9559** | **0.0512** | **+1.68 dB** |
| Grid4D | Cat | Monocular | 29.71 | 0.9640 | 0.0539 | |
| | | **Multi-view** | **31.03** | **0.9716** | **0.0460** | **+1.32 dB** |
| | Spiderman | Monocular | 25.37 | 0.9453 | 0.0590 | |
| | | **Multi-view** | **30.24** | **0.9712** | **0.0400** | **+4.87 dB** |
| | Toy | Monocular | 31.29 | 0.9639 | 0.0312 | |
| | | **Multi-view** | **33.18** | **0.9630** | **0.0377** | **+1.89 dB** |

## G COMPUTATIONAL EFFICIENCY ANALYSIS

We benchmarked the resource requirements (VRAM and training time) of our method against base-lines on a single NVIDIA RTX 4090 GPU. We utilized the *Cat*, *Spiderman*, and *Toy* sequences from the *SynthoMotion-360* dataset, which feature diverse motion complexities and sequence durations (80 to 260 frames).

Table A7 summarizes the peak VRAM usage and total training time for all methods.

**Memory Consumption (VRAM).** Our method exhibits a higher memory footprint (19.7 GB) com-pared to baseline methods. This is a design choice inherent to our **Parametric SDF** representation. Unlike deformation-based methods that store a single canonical template, our method explicitly stores polynomial and Fourier basis coefficients for every vertex in a dense 3D grid. While memory-intensive, this storage strategy is essential for handling **topological changes** and enforcing strict **temporal coherence**.

**Training Efficiency.** In terms of training duration, our method (121 min) demonstrates competitive efficiency. It is **significantly faster** ($1.5 \times -2.1\times$) than complex optimization-based methods like **Dynamic 2DGS** and **SC-GS**. While slower than some lightweight baselines, the additional time is

Table A7: **Computational Efficiency Comparison.** We report the peak VRAM usage and total training time.

| Method | Peak VRAM (GB) | Training Time (min) |
|---|---|---|
| **Deformable 3DGS** | 1.2 | 55.33 |
| **Grid4D** | 3.2 | 62.31 |
| **NeuS2** | 6.2 | 72.31 |
| **DG-Mesh** | 9.3 | 85.61 |
| **AT-GS** | 3.1 | 93.21 |
| **Ours (p-SDF)** | **19.7** | **121.02** |
| **Dynamic 2DGS** | 2.4 | 184.01 |
| **SC-GS** | 1.8 | 253.67 |

invested in differentiable iso-surfacing and PBR material estimation, which we believe is a justified cost for the superior geometric quality.

## H   ANALYSIS OF P-SDF UNDER MONOCULAR CONSTRAINTS

To further clarify the distinction between our multi-view framework and prior monocular-based approaches, and to investigate the necessity of multi-view observations for our representation, we evaluated our method on two additional datasets under a strict monocular setting: the **DG-Mesh dataset** (Liu et al., 2024) and the **D-NeRF dataset** (Pumarola et al., 2021a).

In this experiment, we adapted our framework to train under a monocular setting. Unlike our main experimental configuration, where gradients from **multiple fixed camera views** constrain the geometry simultaneously at each timestamp, this setting is restricted to **a single, randomly selected viewing angle per timestamp**. Consequently, the optimization relies solely on photometric consistency from this solitary view at any given moment.

### H.1   RESULTS AND ANALYSIS

We present the qualitative reconstruction results in Figure A8, visualizing both Novel View Synthesis (NVS) and the underlying geometry (Normal Maps) across three timestamps.

**Performance on DG-Mesh (Slow Dynamics).**   As shown in Figure A8 (Top), the rendered NVS results remain visually coherent. This relatively reasonable performance is attributed to the dataset's characteristics, which feature slow and smooth walking motions. However, a closer inspection of the **Normal Maps** reveals that the reconstructed surfaces exhibit noticeable high-frequency noise and "bumpiness" (e.g., on the hair and clothing). Unlike the smooth surfaces obtained in our multi-view experiments, the monocular constraint is insufficient to regularize the surface details fully.

**Performance on D-NeRF (Large Deformations).**   On the D-NeRF dataset (Figure A8, Bottom), the reconstruction quality degrades significantly. This sequence involves large-scale non-rigid deformations (jumping). As observed in the Normal Maps, the geometry suffers from severe distortions; limbs appear disconnected or abnormally thin (see $T_0$), and the torso exhibits blocky artifacts ($T_1$). The combination of large motion and sparse spatial information prevents the p-SDF from finding a valid geometric trajectory that satisfies the changing monocular views, leading to a breakdown in topological consistency.

### H.2   DISCUSSION

In our p-SDF framework, the geometry is represented by parametric curves stored at grid nodes. In a multi-view setting, the intersection of camera rays from different angles provides a strong supervisory signal that constrains the *curve values* (i.e., the SDF evolution) at each timestamp. In the monocular setting, this multi-view constraint is absent.

However, the impact of this absence varies with motion magnitude. For s**low dynamics** (e.g., DG-Mesh), the inherent temporal continuity of our p-SDF representation acts as a strong regularizer—

similar to its effectiveness in temporal interpolation—allowing the model to maintain a plausible global structure even with sparse spatial constraints. Conversely, under **large-scale motions** (e.g., D-NeRF), this regularization is insufficient, the supervision signal for the curve values becomes incomplete, leading to *shape-radiance ambiguity*. The optimization process struggles to distinguish between geometry changes and appearance changes based on a single view, resulting in the observed noise and geometric degradation.

This analysis confirms that while p-SDF is a powerful representation for continuous dynamics, explicit multi-view consistency is essential for achieving the high-fidelity, artifact-free surface reconstruction demonstrated in our main paper.

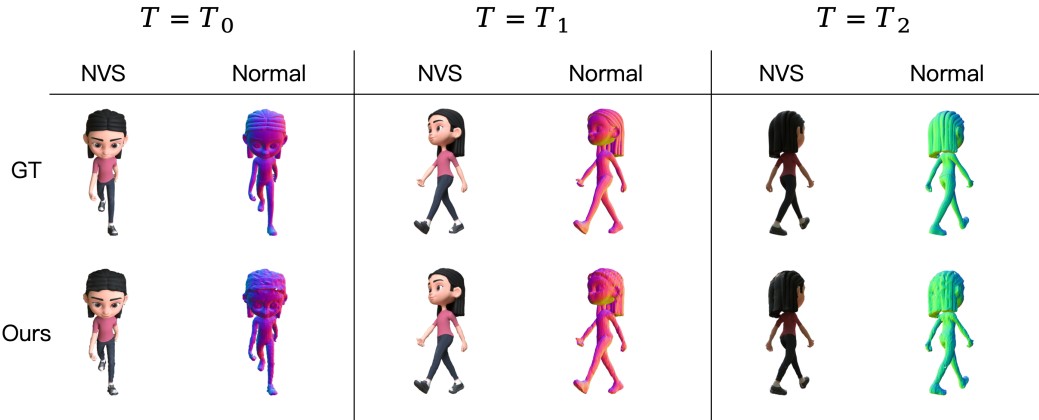

(a) Results on DG-Mesh (Monocular Input)

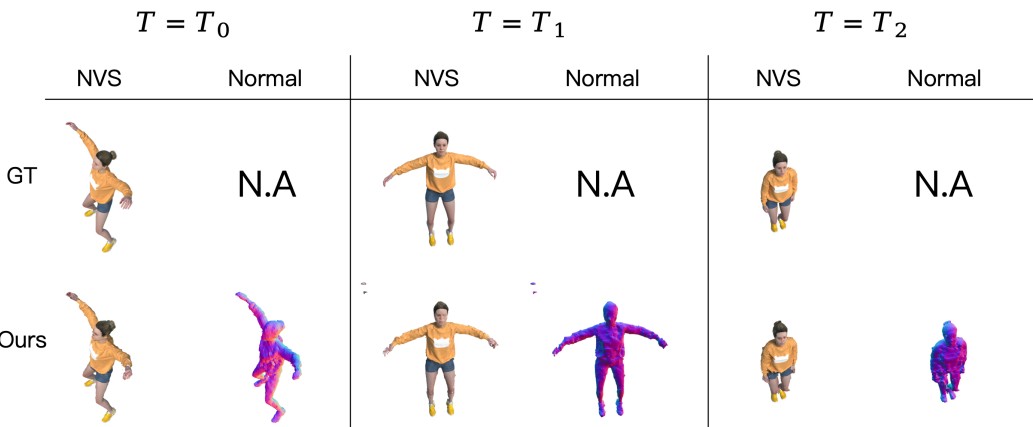

(b) Results on D-NeRF (Monocular Input)

Figure A8: **Qualitative evaluation under the Monocular Setting. (a) DG-Mesh:** Due to slower motion, the method recovers a plausible global shape, but the Normal maps exhibit high-frequency noise and lack surface smoothness compared to multi-view results. **(b) D-NeRF:** Due to large-scale deformation and sparse views, the method suffers from shape-radiance ambiguity, resulting in severe geometric distortions (broken limbs, blocky surfaces) and floating artifacts in NVS.

## J  STATEMENT ON THE USE OF LLMS

Pursuant to the conference policy, the authors wish to disclose the use of a Large Language Model (LLM) as an assistive tool in the preparation of this paper. We utilized Google Gemini for the purpose of language editing and refinement. The scope of its use was strictly confined to improving the manuscript's readability, including tasks such as grammar correction, sentence restructuring for clarity, and style enhancements. We state that the LLM was not involved in the ideation of the research, the formulation of the methodology, the generation or analysis of experimental results, or the drawing of scientific conclusions.

