# OpenReview forum: "Parametric SDF for Dynamic Surface Reconstruction"
_ICLR.cc/2026/Conference — Submitted to ICLR 2026_

### Official Review · Reviewer_1jmj · 2025-10-24

**Soundness:** 3
**Presentation:** 3
**Contribution:** 3
**Rating:** 8
**Confidence:** 4

**Summary:**

This paper proposes a novel framework for dynamic surface reconstruction based on a parametric Signed Distance Function (p-SDF) representation. The core idea is to generalize static SDF fields—where each spatial point stores a single constant value—into time-dependent parametric curves, with each curve modeling the continuous temporal evolution of the SDF at that point. This formulation enforces temporal coherence while accommodating large deformations and topological changes. At any given timestamp, a static SDF field can be queried from the parametric function and converted into an explicit mesh through differentiable iso-surfacing. Combined with a physically based differentiable renderer and a dynamic tri-plane appearance field, the framework enables end-to-end optimization from multi-view video observations. Experiments on synthetic and real-world datasets demonstrate state-of-the-art geometric accuracy, temporal consistency, and visual fidelity. The proposed method demonstrates smooth, temporally stable reconstructions that surpass prior dynamic scene methods in both quality and robustness.

**Strengths:**

1. The introduction of parametric SDF trajectories is novel and well-explored in the paper. Modeling the evolution of SDF values as continuous parametric curves provides an intuitive and principled way to ensure temporal coherence while still allowing for topological flexibility.
2. The proposed pipeline combines p-SDF-based geometry, differentiable iso-surfacing, and physically based rendering in a unified system. The design practically bridges geometric modeling and appearance estimation with end-to-end differentiability.
3. The method consistently outperforms previous approaches in both quantitative metrics (PSNR, Chamfer distance, MAE) and qualitative visual quality. The results on highly dynamic and complex datasets validate the robustness of the approach.

**Weaknesses:**

1. Since each spatial point stores its own set of curve parameters, the memory footprint may increase rapidly with larger scenes or longer sequences. The paper would benefit from a more detailed discussion on scalability. In addition, it is recommended to include a table comparing the computational requirements and efficiency (e.g., training time and inference time) of different methods to better illustrate the practicality of the proposed approach.
2. The framework assumes synchronized multi-view inputs, which may limit its applicability in monocular or unsynchronized video settings.
3. The paper could include a detailed analysis or ablation on how the choice and balance of polynomial versus Fourier basis functions affect reconstruction quality, especially for non-periodic motion.

**Questions:**

1. How do the memory consumption and runtime scale with longer sequences (e.g., several thousand frames)? It would be helpful to understand the computational trade-offs and whether the method remains efficient for large-scale dynamic scenes.
2. Can the proposed p-SDF representation handle discontinuous or abrupt temporal changes, such as object appearance or disappearance?

---

> ### Author Response · Authors · 2025-11-27
>
> We sincerely thank the reviewer for the positive assessment and for recognizing the "novelty" and "robustness" of our p-SDF framework. We appreciate your constructive suggestions regarding scalability and implementation details, which have driven significant improvements in our revision.
>
> ---
>
> ### **1. Scalability and Efficiency**
>
> We appreciate the reviewer's focus on practicality. We have addressed this from three angles:
> 1. On Training Efficiency (Table A7): We provide a transparent comparison in Appendix G. We acknowledge that our method (~121 min) requires more training time than lightweight deformation-based Gaussian methods (e.g., Deformable 3DGS, ~55 min). This is a necessary **Quality-Efficiency Trade-off**: unlike Gaussian methods that only optimize point clouds for rendering, our framework solves a more complex inverse problem—extracting watertight meshes via differentiable iso-surfacing and decomposing PBR materials. However, we are significantly faster than optimization-heavy methods like Dynamic 2DGS (184 min) and SC-GS (253 min). We believe the 2-hour training time is a reasonable cost for achieving state-of-the-art geometric fidelity and relightability.
>
> | **Method** | **Peak VRAM (GB)** | **Training Time (minute)** |
> | --- | --- | --- |
> | **Deformable 3DGS** | 1.2 | 55.33 |
> | **Grid4D** | 3.2 | 62.31 |
> | **NeuS2** | 6.2 | 72.31 |
> | **DG-Mesh** | 9.3 | 85.61 |
> | **AT-GS** | 3.1 | 93.21 |
> | **Ours (p-SDF)** | **19.7** | **121.02** |
> | **Dynamic 2DGS** | 2.4 | 184.01 |
> | **SC-GS** | 1.8 | 253.67 |
>
> 2. Theoretical Scaling: The memory footprint is governed by the grid resolution N and basis counts K: $M_{geo} \propto O(N^3 \times K)$ . While cubic growth is a constraint, our ablation in Table A4 shows that a resolution of $96^3$ is sufficient for high-fidelity reconstruction, striking an optimal balance between quality and cost.
> 3. Scaling to Long Sequences (Thousands of Frames): For extremely long sequences, fitting a single set of global basis functions is indeed challenging. To scale effectively without unbounded memory growth, our framework can employ a Temporal Sliding Window strategy—processing the video in overlapping temporal blocks. This ensures memory usage for any optimization step remains constant (bounded by the window size) regardless of the total video duration.
>
> ---
>
> ### **2. Monocular vs. Multi-view Applicability**
> We acknowledge this limitation but emphasize it as a deliberate design choice. Our goal with p-SDF is to solve the "ill-posed" geometry problem to achieve state-of-the-art geometric accuracy, which fundamentally necessitates the strong constraints provided by multi-view observations.
>
> ---
>
> ### **3. Ablation on Basis Functions**
>
> We have added a ablation study in Appendix D.1 (Table A3) to validate our hybrid design.
>
> | **Method** | **PSNR↑** | **SSIM↑** | **LPIPS↓** | **MAE↓** | **Chamfer↓** |
> | --- | --- | --- | --- | --- | --- |
> | (a) Polynomial Only | 25.910 | 0.952 | 0.049 | 2.492 | 2.438 |
> | (b) Fourier Only | 35.904 | 0.978 | 0.012 | 0.505 | 0.086 |
> | **Ours (Full Model)** | **36.337** | **0.980** | **0.011** | **0.465** | **0.084** |
>
> * Polynomial Only: Suffers a catastrophic drop in performance (PSNR drops by >10 dB), this confirms that low-degree polynomials fail to capture high-frequency dynamics.
> * Fourier Only: Performs strongly (PSNR 35.90 dB), proving frequency terms are the primary driver for detail.
> * Hybrid (Full Model): Combines the best of both worlds, yielding the highest accuracy (PSNR 36.34 dB). The polynomial terms effectively model the low-frequency motion, allowing Fourier terms to focus on refining details.
>
> ---
>
> ### **4. Handling Discontinuous/Abrupt Changes**
>
> This is an excellent theoretical question. We have added a dedicated analysis in Appendix A (Modeling Topological Changes) to answer this.
>
> * Signal Fitting (Figure A1): We show that our basis functions can successfully fit a Ground Truth SDF signal containing a sharp, step-like drop (representing a topological event).
> * Reconstruction (Figure A2): We tested the pipeline on a synthetic "Torus-to-Sphere" sequence. As shown in *New Rebuttal Video*, our method successfully reconstructed the hole closing smoothly over time, using only image supervision.
>
> This confirms that p-SDF is robust to abrupt topological changes.
>
> ---
>
> ### **Summary**
>
> We sincerely thank the reviewer for the positive evaluation and insightful feedback. We believe the newly added efficiency benchmarks (Appendix G), basis function ablations (Appendix D), and topological analysis (Appendix A) have significantly strengthened the manuscript and fully addressed the questions raised regarding scalability and robustness.

---

> > ### Comment · Reviewer_1jmj · 2025-11-27
> >
> > Thanks for the rebuttal. I appreciate the authors for the comments and all my concerns are addressed. The results of the topological analysis are impressive and please include all the discussions in the revised paper. I would maintain my positive rating and good luck.

---

> > > ### Author Response · Authors · 2025-11-27
> > > **Response to Reviewer 1jmj**
> > >
> > > We sincerely thank Reviewer 1jmj for the continued support and positive assessment.
> > >
> > > We are delighted to hear that our analysis on topological changes addressed your concerns effectively. As per your suggestion, we have already incorporated these detailed discussions and visualizations into Appendix A of the revised manuscript. We will ensure that these analyses, along with the efficiency comparisons and basis function ablations, remain a core part of the final version of the paper.
> > >
> > > We deeply appreciate your constructive feedback, which has significantly strengthened the completeness and robustness of our work.

---

> > > > ### Comment · Reviewer_1jmj · 2025-11-28
> > > >
> > > > Thank you for the response. As a minor comment, I would suggest including reconstruction results (either quantitative or qualitative) for monocular inputs using the proposed method in the paper. From my perspective, this would help address concern regarding the comparison between monocular and multi-view results.

---

> > > > > ### Author Response · Authors · 2025-11-28
> > > > > **Response to Reviewer 1jmj**
> > > > >
> > > > > We thank the reviewer for this insightful suggestion. We agree that including an analysis of our method under monocular settings would be very valuable to further clarify the distinction between monocular and multi-view results.
> > > > >
> > > > > As noted in our Limitations, our current framework relies on multi-view consistency to resolve geometric ambiguities. Applying our p-SDF directly to monocular input without additional priors typically results in degraded geometric fidelity. We will gladly include a qualitative demonstration of this in the **Appendix**. This visualization will highlight precisely why multi-view observations are essential for the high-fidelity surface reconstruction we aim to achieve.
> > > > >
> > > > > Thank you again for helping us strengthen the positioning of our work.

---

> > > > > ### Author Response · Authors · 2025-12-03
> > > > > **Response to Reviewer 1jmj**
> > > > >
> > > > > We sincerely thank the reviewer for this insightful suggestion. We are pleased to report that we have conducted this exact experiment and included the results in the newly added *Appendix H* and *Figure A8* of the revised paper.
> > > > >
> > > > > Specifically, we adapted our framework to optimize against only a single view per timestamp on the **DG-Mesh** and **D-NeRF** datasets. Our findings (detailed in *Section H.1* ) are as follows:
> > > > > 1. **Slow Dynamics (DG-Mesh)**: Our p-SDF maintains global coherence but suffers from high-frequency surface noise due to the lack of multi-view constraints.
> > > > > 2. **Large Deformations (D-NeRF)**: The reconstruction degrades significantly (e.g., broken limbs), confirming that single-view supervision is insufficient to resolve the shape-radiance ambiguity under large non-rigid motions.
> > > > >
> > > > > Conclusion: This analysis explicitly validates our core motivation: while p-SDF provides strong temporal regularization, **explicit multi-view consistency** is essential for achieving the artifact-free, high-fidelity geometry we target.
> > > > >
> > > > > Thank you again for helping us strengthen the comprehensiveness of our work.

---

### Official Review · Reviewer_Bb8o · 2025-10-29

**Soundness:** 1
**Presentation:** 1
**Contribution:** 2
**Rating:** 2
**Confidence:** 4

**Summary:**

The paper presents a method for reconstructing dynamic scenes from multi-view videos. The central goal of the work is to recover accurate, temporally coherent geometry, and the key idea proposed to achieve this is a temporally parameterised SDF representation. The achieved reconstruction is evaluated on multiple datasets (the newly proposed SynthoMotion-360 dataset and the established CMU Panoptic Studio) and compared with multiple dynamic novel-view synthesis and geometry reconstruction methods, achieving lower geometry and novel-view errors. In addition, the work estimates materials and lighting in a disentangled way and shows some relighting examples.

**Strengths:**

1. The idea of parametric SDF that traces a continuous SDF trajectory, with a basis formulation that inherently enforces temporal coherency, is well-motivated and well-formulated.
2. The combination of polynomial and Fourier bases that respectively capture both large-scale deformations and fine-grained high-frequency motions is also a nice takeaway and well-evaluated.
3. Generally, the paper is clearly written and structured.

**Weaknesses:**

Despite some interesting insights and its originality in parametric SDF representation, I think the current submission is not ready for publication. The primary reason is the misleading positioning of the proposed method with respect to the related works and its unfair comparison with the state of the art. Please see the following points for further details:

1.	While the proposed method performs reconstruction from multi-view videos, all the compared methods are designed for reconstruction from monocular videos. This list includes Dynamic-2DGS (Zhang et al., 2024), DGMesh (Liu et al., 2024), Deformable-3DGS (Yang et al., 2024), SC-GS (Huang et al., 2023), and Grid4D (Xu et al., 2024).  Such a comparison is completely unfair. Moreover, the proposed paper doesn't even mention whether these methods are evaluated using a multi-view loss. Note that the proposed method states in the limitations that it doesn’t work in a monocular setting. Then, it is highly unclear why the proposed method is better; is it the new parametric SDF representation or the additional input views?

2.	There are works which reconstruct surfaces from multi-view video, e.g, NeuS2 [Wang et al. 2023], Chen et al. 2025, Zheng et al. 2025. Kindly make such an appropriate comparison.

[Wang et al. 2023] Wang, Yiming, et al. "Neus2: Fast learning of neural implicit surfaces for multi-view reconstruction." Proceedings of the IEEE/CVF International Conference on Computer Vision. 2023.

[Chen et al. 2025] Chen, Decai, et al. "Adaptive and temporally consistent gaussian surfels for multi-view dynamic reconstruction." 2025 IEEE/CVF Winter Conference on Applications of Computer Vision (WACV). IEEE, 2025.

[Zheng et al. 2025] Zheng, Chengwei, et al. "GauSTAR: Gaussian Surface Tracking and Reconstruction." Proceedings of the Computer Vision and Pattern Recognition Conference. 2025.

3.	In the related works, monocular and multi-view works are not clearly differentiated in their approach to dynamics. Moreover, although the proposed work takes multi-view video as input, most discussions in the related work focus on monocular dynamic NeRF/3DGS and are written to imply that the proposed work falls into this category.

4.	The central claim and goal of the proposed method is to achieve higher temporal coherence. I checked the supplemental video, and I do not see any better temporal consistency for the proposed method compared to the other works (eg, for cactus, the results of competing methods are pretty good for dynamic parts). So, my observation conflicts with the main claims of the paper: prior work produces noisy or flickering reconstructions under complex dynamics or large motions, but the proposed work doesn’t. In addition, temporal coherence is not quantitatively measured, as PSNR/MAE do not directly reflect that (especially in the case of multi-view reconstruction, where sufficient supervision is available per frame)

5.	In Section 4.3, it is mentioned, “our method is the first to achieve dynamic reconstructions with precise PBR materials to support realistic dynamic relighting.” This statement seems incorrect; such a task has been addressed before, e.g. see SAFT [Stotko et al.]. Moreover, material/ lighting estimation is not discussed at all in the related works.

[Stotko et al.] Stotko, David, and Reinhard Klein. "SAFT: Shape and Appearance of Fabrics from Template via Differentiable Physical Simulations from Monocular Video." Proceedings of the IEEE/CVF International Conference on Computer Vision. 2025.

Minor comments

1.	Some typos – frist at L159 and reconsturction at L168, and overviews at L200

2.	Referencing grid entry as a “node” might be clearer than calling it a vertex (e.g. L200), as a vertex could be confused with a mesh vertex (which is used during the next stage of mesh optimisation).

3.	L212 - Following prior work on temporal signal modelling, could you please add citations as to which ones?

4. I got the idea behind Fig. 2, but the location of points and SDF vs. time plots do not seem to match exactly. Please double-check.

**Questions:**

Please check the weakness section to find my questions and comments.

---

> ### Author Response · Authors · 2025-11-27
>
> We sincerely thank Reviewer Bb80 for the detailed feedback and for highlighting our parametric SDF formulation as "well-motivated" and "well-formulated". While the major concern raised by the reviewer is "the misleading positioning of the proposed method with respect to the related works and its unfair comparison with the state of the art", would like to provide detailed explanations to clarify this.
>
> ---
>
> ### **1. Concerns about Fairness (Multi-view vs. Monocular)**
>
> We thank the reviewer for raising this point and agree that a fair comparison across different approaches is critical. We sincerely invite the reviewer to look into the following clarification on experimental setup and the motivation behind our proposed multi-view framework:
>
> 1.  **Clarification on Baseline Evaluation:** The reviewer noted a lack of clarity regarding the training setting of the baselines. We confirm that **all baseline methods were optimized in a multi-view setting, utilizing the exact same inputs as our method.**
>
>     * We appreciate the reviewer's constructive suggestion and acknowledge that this detail was not explicitly emphasized in the initial submission. We have revised Section 4.1 to explicitly state that *"methods that are originally designed for monocular settings have been adapted to multi-view for fair comparisons"* to prevent any ambiguity regarding the input protocols.
>
> 1.  **Validity of Comparing Monocular Architectures in Multi-view Settings:** A concern may remain that comparing methods originally designed for monocular settings (e.g., Dynamic-2DGS, Grid4D, DGMesh) against a native multi-view method is unfair. We argue that this comparison is essential to *demonstrate the limitations of current monocular methods when applied to complex data, and to highlight the necessity of explicit multi-view constraints for dynamic surface reconstruction.* Specifically:
>
>     * **Performance of Monocular Architectures in Multi-view Settings:** We conducted a controlled experiment (Table A6) to verify if simply feeding multi-view data to monocular baselines solves the problem.
>       * Significant Quantitative Gains: We confirm that baselines do benefit significantly from multi-view supervision. For instance, Grid4D sees a PSNR improvement of +4.87 dB on the Spiderman scene.
>
>     * **However, their performance gains from multi-view data are limited.** Even when trained with multi-view inputs, these methods struggle because they enforce appearance consistency without enforcing temporal coherence on the underlying geometry. This results in "sparkling" artifacts, broken topology, and temporal jitter under large motions, as shown in *Figure 6, 7, A4, A5 and New Rebuttal Video*. Therefore, a specialized multi-view approach is required for complex real-world scenes.
>     * **Our goal is to establish a new standard for multi-view performance.** We do not claim our method is a monocular solution. Instead, we demonstrate that simply feeding more views to existing monocular algorithms is insufficient. To achieve high-fidelity surface reconstruction in real-world scenarios, a dedicated multi-view architecture is necessary.
>
> As a result, we believe our experiments represent a fair and necessary conclusion: *in complex real-world scenarios, our method benefits from its novel parametric SDF representation, yielding high-quality surfaces, whereas baselines (whether monocular or extended to multi-view) consistently underperform.*
>
> **To better validate our claim, we have revised the sections of geometry quality evaluation and reorganized them into *Section 4.3***, where we provide a detailed discussion on this point. We hope this clarifies our evaluation protocol and the specific gap our work aims to fill. We have updated the manuscript to clearly differentiate between method capabilities and evaluation protocols.

---

> > ### Author Response · Authors · 2025-11-27
> >
> > ### **2. Comparison with Multi-view Dynamic Reconstruction Methods**
> >
> > We thank the reviewer for suggesting these multi-view baselines. We agree that including them substantially strengthens our evaluation and have conducted the comparisons (also included in *Section 4.3, Table 2*):
> >
> > | Method          | PSNR $\uparrow$ | CD ($10^{-3}$) $\downarrow$ | F1 $\uparrow$  | ECD ($10^{-2}$) $\downarrow$ | EF1 $\uparrow$    |
> > |-----------------|------------------|------------------|-------|-------------------|----------|
> > | Deformable-3DGS | 28.895            | 4.535            | 0.663 | 5.038             | 0.4312   |
> > | SC-GS           | 29.881            | 3.051            | 0.770 | 3.898             | 0.5110   |
> > | Dynamic-2DGS    | 29.489           | 2.850            | 0.754 | 3.802             | 0.4939   |
> > | Grid4D          | 30.048            | 4.637            | 0.700 | 4.708             | 0.4604   |
> > | DG-Mesh         | 25.719            | 3.760            | 0.661 | 4.740             | 0.4382   |
> > | **NeuS2**           | 30.070            | 2.253           | **0.908** | 3.220             | 0.5928   |
> > | **AT-GS**           | **32.671**            | 2.339           | 0.865 | 3.227             | 0.5662   |
> > | **Ours**            | 31.653            | **2.220**           | 0.889 | **3.170**             | **0.5955**   |
> >
> > * **NeuS2 [Wang et al. 2023]:** While NeuS2 produces reasonable geometry, it relies on incremental reconstruction that inheretly lacks motion modeling, thereby failing to naturally support time interpolation or achieve temporal smoothness. In contrast, our p-SDF ensures both geometric fidelity and temporal coherence.
> > * **ATGS [Chen et al. 2025]:** ATGS **leverages RAFT [Teed & Jia, 2020] as a motion prior**, although it achieves better psnr, we found it struggles with the large motions in the SynthoMotion360 dataset. Even without the strong RAFT prior, our method achieves superior geometric quality with comparable NVS performance.
> > * **GauSTAR [Zheng et al. 2025]:** We did not include GauSTAR as it strictly requires depth map inputs, rendering it inapplicable to our RGB-only setting.
> >
> > > *[Wang et al. 2023] Wang, Yiming, et al. "Neus2: Fast learning of neural implicit surfaces for multi-view reconstruction." ICCV 2023.*
> > >
> > > *[Chen et al. 2025] Chen, Decai, et al. "Adaptive and temporally consistent gaussian surfels for multi-view dynamic reconstruction." WACV 2025.*
> > >
> > > *[Zheng et al. 2025] Zheng, Chengwei, et al. "GauSTAR: Gaussian Surface Tracking and Reconstruction." CVPR 2025.*
> > >
> > > *[Teed & Jia. 2020] Teed, Zachary, and Jia Deng. "Raft: Recurrent all-pairs field transforms for optical flow." ECCV 2020.*
> >
> > ---
> >
> > ### **3. Related Work Positioning (Multi-view vs. Monocular)**
> >
> > We acknowledge the reviewer's concern that our initial positioning may have caused confusion. We strictly clarify that we do not claim to be a monocular method. To resolve this ambiguity, we have thoroughly revised *Section 2* to clearly categorize methods into **Monocular** (e.g., D-NeRF, Dynamic-2DGS) and **Multi-view** (e.g., NeuS2).
> >
> > And we now explicitly position baselines as *"originally designed for monocular settings"* in experiments *(Table 2 and 3)*, highlighting that unlike these methods relying on learned deformation priors, our method leverages explicit multi-view geometric consistency to achieve high-fidelity reconstruction.
> >
> > ---
> >
> > ### **4. Significance on Temporal Coherence**
> >
> > We appreciate this observation. The temporal coherence gap was less apparent in the "Cactus" supplementary video because that scene features simple topology and limited motion. We have addressed this by providing more challenging visual evidence and quantitative metrics:
> >
> > * **Complex Scenarios:** As shown in *Figure 6 and New Rebuttal Video*, baselines exhibit severe flickering and broken geometry under *large deformations* and *complex topological changes*, whereas p-SDF remains robust.
> > * **Quantitative Evidence:** To validate this quantitatively, we report a *Temporal Coherence Metric* in *Table 4 (Section 4.3)*. Our method significantly outperforms the baselines on this metric.
> >
> > ---
> >
> > ### **5. Clarification of the PBR / Relighting Claim**
> >
> > We thank the reviewer for pointing this out. We have cited this work and refined our claims to be more precise:
> >
> > * We have removed *"our method is the first to achieve dynamic reconstructions with precise PBR materials"* statement in *Section 4.4*.
> > * We have included SAFT in the related work section. SAFT relies on a *pre-scanned geometry template*, limiting it to objects with fixed topology (e.g., fabrics). In contrast, our method is *template-free*. We simultaneously reconstruct dynamic geometry, *topological changes* (which SAFT cannot handle), and PBR materials from scratch.

---

> > > ### Author Response · Authors · 2025-11-27
> > >
> > > ### **6. Minor Comments on Typos, Citations, and Presentation**
> > >
> > > We thank the reviewer for the detailed proofreading. We have implemented all corrections:
> > >
> > > * **Typos & Terminology:** All typos have been corrected. We now consistently use the term **"node"** for grid entries to distinguish them from mesh vertices.
> > > * **Citations:** We added the requested citations regarding temporal signal modeling (e.g., Gaussian flow) at Line 212.
> > > * **Figure 2:** We have corrected the visual correspondence between point locations and SDF plots in Figure 2.
> > >
> > > ---
> > >
> > > ### **Summary**
> > >
> > > We appreciate the reviewer’s constructive criticism. We believe the additional experiments and clarifications regarding fairness and evaluation protocols have fully addressed the concerns raised.

---

### Official Review · Reviewer_2jbt · 2025-10-31

**Soundness:** 3
**Presentation:** 2
**Contribution:** 3
**Rating:** 8
**Confidence:** 3

**Summary:**

The paper introduced a parametric SDF representation for dynamic 3d scenes. My understanding is that every frame of the dynamic scene is represented by an SDF. Rendering this dynamic scene at any given timestamp, which differs from the timestamps at which the scene was captured in every frame, requires proper capture and representation of the scene's dynamics. I am not sure if this is the main motivation, but this is my assumption, based on my older experience on the topic. Here, it is clear that the authors do not propose tracking
of the dynamics of the moving objects in the scene, but rather how to properly interpolate motion between some of the frames. Equally, they aim to evaluate how their proposed representation can be utilized for novel view synthesis of dynamic scenes. So, parametric SDF proposes representing a dynamic scene as an SDF evolving over time, where for every SDF voxel, there is a parametric function that represents the change in SDF values over time. This was used for capturing geometry. The appearance has been captured in three orthogonal planes, representing dynamic changes in the appreciation for every SDF voxel.

**Strengths:**

I find the two key ideas, parametric SDF and appearance tri-plane representation, quite interesting. They seem quite simple, but effective. Optimizing the decoded appearance with PBR rendering in an end-to-end fashion allows for the well-adjustment of the SDF basis function parameters to properly interpolate the geometry. Additionally, learning well, the appearance decomposed in the material properties, aligned with the actual multiple images, allows for the learning of decoupled scene appearance, enabling effective scene relighting. I find these elements as a strong contribution to this particular problem.

**Weaknesses:**

Although the technical part of the paper is well described and comprehensive, I still felt that the exact motivation was missing. Simply said, describe what your input is in every frame, and what you want to obtain as a result. If I haven't misunderstood, rendering the scenes in any time frame other than those where the multi-view capture occurred is the key.

**Questions:**

In the experimental results, it is stated that a larger number of views was used for training and a smaller number for testing, both for NVS and for geometric errors.
For NVS, I can understand that the scenes are rendered in the test views and corresponding measures have been evaluated. However, if the parametric SDF has been optimized on the larger
number of views(training), what does the testing in terms of the geometric accuracy with the smaller number of views mean?
If my understanding is correct, the learned parametric SDF serves for interpolating the dynamic scene at new intermediate timestamps. So I would expect learning from N views and
for K frames out of the total M frames. Now, for the geometric accuracy, I would check the geometric error between the mesh produced from the parametric SDF at the M-K test frames not used for the learning. I would really love to understand this and strongly encourage you to explain it in the rebuttal

---

> ### Author Response · Authors · 2025-11-27
>
> We sincerely thank the reviewer for the strong support and for recognizing our Parametric SDF and Appearance Tri-plane as "simple but effective" and a "strong contribution" to the field. We appreciate the opportunity to clarify our motivation and experimental protocol.
>
> ---
>
> ### **1. Clarification on Motivation (Input & Output)**
>
> The reviewer’s intuition is excellent. While our continuous p-SDF representation *naturally* supports temporal interpolation (rendering at unobserved timestamps), our primary research goal aligns with the **standard Dynamic Surface Reconstruction task**. We clarify the pipeline below:
>
> *  **Input:** A synchronized multi-view video sequence (RGB images + camera poses) covering the full duration of the event.
> *  **Output:** A continuous 4D spatio-temporal representation (comprising the p-SDF geometry and dynamic appearance field).
> *  **Primary Goal:** To solve the inverse problem of recovering high-fidelity, topologically consistent 3D geometry from 2D observations and synthesizing novel views for the observed frames.
>
> ---
>
> ### **2. Explanation of Experimental Protocol (Spatial vs. Temporal Split)**
>
> We Thank the reviewer posed an insightful question regarding our evaluation setup: "What does testing geometric accuracy mean if trained on all frames?"
>
> * Following standard benchmarks in the field (e.g., NeuS2, Dynamic-3DGS), our main quantitative results (Tables 1, 2, 3) utilize a Spatial Hold-out Protocol. We use all timestamps for training to capture the full temporal evolution, but we hold out specific Camera Views for evaluation.
>
> ### **3. New Experiment: Temporal Interpolation (Appendix E)**
>
> We appreciate your strong encouragement to explore this direction. As you correctly pointed out, the continuous nature of p-SDF should theoretically enable high-quality motion interpolation. To validate your hypothesis, we conducted the exact experiment you described and added the full results to Appendix E (Table A5 & Figure A7).
>
> | **Scene** | **Split** | **CD (↓)** | **PSNR (↑)** | **Normal MAE (↓)** |
> | --- | --- | --- | --- | --- |
> | **Toy** | Trained Frames (Even) | $8.68 \times 10^{-5}$ | 36.07 | 0.52 |
> |  | **Interpolated Frames (Odd)** | $\mathbf{9.68 \times 10^{-5}}$ | **32.90** | **0.66** |
> | **Spiderman** | Trained Frames (Even) | $3.78 \times 10^{-5}$ | 30.00 | 0.98 |
> |  | **Interpolated Frames (Odd)** | $6.55 \times 10^{-3}$ | 18.90 | 4.46 |
>
> * **Protocol:** We employed a **Temporal Hold-out Split** on the SynthoMotion-360 dataset. We trained the model using only **even frames ( t=0, 2, 4...  )** and evaluated geometry and NVS on unseen **odd frames** (t=1, 3, 5...).
> * Results and Limitations (Table A5 & Fig. A7):
>   * **Effective Interpolation for Smooth Motion**: In the "Toy" scene, our p-SDF successfully captures the continuous trajectory, yielding low geometric error on unseen frames (CD: $9.68 \times 10^{-5}$) comparable to training frames.
>   * **Limitation under Large Dynamics**: In the "Spiderman" scene, which features rapid, large-scale non-rigid deformations, we observe a performance drop on unseen frames (CD increases to $6.55 \times 10^{-3}$). This degradation occurs because the low-rank polynomial and Fourier basis functions may underfit extremely high-frequency temporal variations when sampling is sparse.
>   * **Conclusion**: This validates that p-SDF acts as a strong temporal regularizer, enforcing smoothness. While highly effective for continuous dynamics, sufficient temporal sampling remains necessary for capturing extreme, fast-changing motions.
>
> ---
>
>
> ### **Summary**
>
> We thank the reviewer for the positive assessment and valuable insights. We hope the clarifications on our experimental protocol have resolved the confusion regarding geometric evaluation. Moreover, the newly added temporal interpolation experiment in Appendix E—conducted specifically at your suggestion—confirms your intuition that p-SDF effectively serves as a continuous temporal regularizer, enabling motion interpolation.

---

### Official Review · Reviewer_XrhY · 2025-11-01

**Soundness:** 3
**Presentation:** 3
**Contribution:** 3
**Rating:** 4
**Confidence:** 3

**Summary:**

This paper proposes a parametric SDF for dynamic surface reconstruction. The parametric SDF is built basd on SDF and temporally varying signal modeling with basis functions. In implementation, this parametric function is modeled by a grid and the appearance is modeled by a 4D hash grid. With differentiable rendering, the parametric SDF and appearance are optimized. In experiments, the work curates a new synthetic benchmark, SynthoMotion-360. The experimental results on SynthoMotion-360, DiVa-360 and CMU Panoptic Studio show that the method achieves promoising results, with better fidelity and coherence.

**Strengths:**

1. The method introduces a parametric SDF for dynamic surface reconstuction, which leverages the advantages of SDF and temporlly varying signal modeling with basis functions.
2. The method distanges geometry and appearance to optimize the parametric SDF and material properties.
3. The work curates a new synthetic benchmark to evaluate the dynamic surface reconstruction.
4. The method achieves promoising results for dynamic surface reconstruction.

**Weaknesses:**

1. The quantitative surface reconsutruction evaluation is only test on the synthetic benchmark.
2. For more complex scenes, like the scenes on CMU Panoptic Studio dataset, the method fails to reconstruct high-fidelity surfaces.
3. The method cannot recover complex geometris and details.

**Questions:**

1. Many works focus on dynamic scene reconstruction and they evaluated their methods on some human datasets. It is better to also evaluate the proposed methods on human dataset, like ZJU-Mocap and PeopleSnapshot.
2. It seems that NeuS2 is a SOTA SDF-based method that can reconstruct dynamic scenes. It is better to compare with this method.
3. Can you discuss training efficiency for different methods, including NeuS2.
4. For the detail reconstruction, is it possible to increase the grid resolution to improve the reconstruciton?
5. Can you show some reconstruciton examples under challenging lighting conditions (Line103)?

---

> ### Author Response · Authors · 2025-11-27
>
> We sincerely thank Reviewer XrhY for their constructive feedback and for recognizing the "promising results" and "fidelity" of our work. We have addressed all specific questions with additional experiments below.
>
> ### **W1. Quantitative Geometric Evaluation on Real Datasets**
>
> We appreciate the reviewer noting the lack of quantitative metrics for real-world data. Actually, that is why we construct a new synthetic dataset (SynthoMotion-360) involving large motion cases (so that ground-truth meshes are available for a comprehensive evaluation on geometry quality). While datasets like CMU Panoptic Studio and DiVa-360 lack ground-truth meshes (making surface normal error impossible to compute), we agree that quantitative evaluation is feasible using the raw scanned point clouds.
>
> Following the reviewer's suggestion, we evaluated the **Chamfer Distance (CD)** and **F1 scores** against scanned point clouds. As shown in *Table 3 (Section 4.3)*, our method achieves state-of-the-art reconstruction accuracy, quantitatively validating the visual improvements. We also present this table as follows:
>
> | Method          | CD ($10^{-3}$) $\downarrow$ | F1 score $\uparrow$  | Aspect>4 (%) $\downarrow$ | Radius>4 (%) $\downarrow$ | MinAngle<10° (%) $\downarrow$ |
> |-----------------|------------------|----------|----------------|----------------|---------------------|
> | Deformable-3DGS | 2.749            | 0.887    | 22.00          | 26.59          | 12.00               |
> | SC-GS           | **2.372**            | 0.893    | 21.93          | 26.51          | 11.97               |
> | Dynamic-2DGS    | 199.6            | 0.763    | 22.66          | 27.40          | 12.44               |
> | Grid4D          | 2.554            | 0.887    | 21.97          | 26.55          | 11.98               |
> | DG-Mesh         | 103.7            | 0.789    | 22.51          | 26.50          | 12.90               |
> | Ours            | 2.375            | **0.907**    | **20.51**          | **24.40**          | **11.70**               |
>
> ### **W2 & W3. Geometry Quality for Complex Cases**
>
> We thank the reviewer for this observation regarding complex scenes (e.g., CMU Panoptic Studio). We acknowledge that these datasets represent an extreme stress test, featuring topological changes and severe inter-person occlusions. regarding this, we would like to clarify:
>
> * **Relative Performance:** While these conditions remain an open challenge for the field, we emphasize that compared to baselines—which suffer from broken geometry and significant noise (*Figure 6*)—our method successfully reconstructs *topologically complete* and *temporally coherent* surfaces that clearly separate interacting individuals. This is also evidenced by our *New Rebuttal Video* in the supplementary materials.
> * **Contribution:** Our primary contribution lies precisely in this improvement. By outperforming existing methods in these "in-the-wild" scenarios, we bridge the gap between synthetic prototypes and real-world applicability.
>
> ### **Q1. Evaluation on Human Dataset**
>
> We appreciate the suggestion. We would like to clarify that our **CMU Panoptic Studio** dataset already includes human data, covering single person motions and multiple person interactions. As shown in *Table 3, Figure 6*, and our *New Rebuttal Video*, our method remains robust, outperforming baselines in both visual fidelity and quantitative metrics.
>
> ### **Q2. Comparison with NeuS2**
>
> We have added a direct comparison with NeuS2 in *Table 2* (*Section 4.3*):
>
> | Method          | PSNR $\uparrow$ | CD ($10^{-3}$) $\downarrow$ | F1 $\uparrow$  | ECD ($10^{-2}$) $\downarrow$ | EF1 $\uparrow$    |
> |-----------------|------------------|------------------|-------|-------------------|----------|
> | Deformable-3DGS | 28.896            | 4.535            | 0.663 | 5.038             | 0.4312   |
> | SC-GS           | 29.881            | 3.051            | 0.770 | 3.898             | 0.5110   |
> | Dynamic-2DGS    | 29.489            | 2.850            | 0.754 | 3.802             | 0.4939   |
> | Grid4D          | 30.047            | 4.637            | 0.700 | 4.708             | 0.4604   |
> | DG-Mesh         | 25.719            | 3.760            | 0.661 | 4.740             | 0.4382   |
> | **NeuS2**           | 30.070            | 2.253           | **0.908** | 3.220             | 0.5928   |
> | **Ours**            | **31.653**            | **2.220**           | 0.889 | **3.170**             | **0.5955**   |
>
> While NeuS2 produces reasonable static geometry, it relies on incremental reconstruction. This approach inherently lacks global motion modeling, resulting in poor temporal smoothness and an inability to interpolate frames. In contrast, our p-SDF ensures both geometric fidelity and temporal coherence.

---

> > ### Author Response · Authors · 2025-11-27
> >
> > ### **Q3. Training Efficiency**
> > On Training Efficiency (Table A7): We provide a transparent comparison in Appendix G. We acknowledge that our method (~121 min) requires more training time than lightweight deformation-based Gaussian methods (e.g., Deformable 3DGS, ~55 min). This is a necessary **Quality-Efficiency Trade-off**: unlike Gaussian methods that only optimize point clouds for rendering, our framework solves a more complex inverse problem—extracting watertight meshes via differentiable iso-surfacing and decomposing PBR materials. However, we are significantly faster than optimization-heavy methods like Dynamic 2DGS (184 min) and SC-GS (253 min). We believe the 2-hour training time is a reasonable cost for achieving state-of-the-art geometric fidelity and relightability.
> >
> > | Method               | Training Time (minute) |
> > |----------------------|---------------------|
> > | Deformable 3DGS      | 55.33               |
> > | Grid4D               | 62.31               |
> > | NeuS2                | 72.31               |
> > | DG-Mesh              | 85.61               |
> > | AT-GS                | 93.21               |
> > | Ours (p-SDF)         | **121.02**              |
> > | Dynamic 2DGS         | 184.01              |
> > | SC-GS                | 253.67              |
> >
> >
> >
> > ### **Q4. Grid Resolution and Detail Reconstruction**
> >
> > We confirm that increasing grid resolution directly enhances geometric detail (*Table A4 in Appendix D.4*).
> >
> > * **Ablation:** We conducted an ablation study across grid resolutions of $64^3$, $96^3$, and $128^3$.
> > * **Trade-off:** As expected, higher resolutions capture finer high-frequency details (e.g., facial features, clothing wrinkles) but increase memory consumption. We selected $96^3$ as our default to optimally balance high-fidelity detail with computational resource requirements.
> >
> > ### **Q5. Handling Challenging Lighting**
> >
> > Thanks for raising this issue. As noted by the reviewer, we have explicitly claimed that "Handling Challenging Lighting" (Line 103 in the original submission). This denotes some non-Lambertian effects at reflective surface. An example is the *Cello* scene in *Figure 6 (Section 4.2)*, where the cello involves reflective surface. With explicit PBR modeling, our methods successfully recover the geometry of the reflective region while other baselines typically fail to handle this.

---

### Author Response · Authors · 2025-11-27

*Dear Reviewers,*

We are encouraged by the positive evaluations from ***Reviewers 2jbt*** and ***1jmj*** regarding the significance of our work, and we sincerely appreciate the constructive feedback from ***Reviewer Bb80*** as well as the detailed suggestions from ***Reviewer XrhY***. Below, we summarize the major revisions made in response to the remaining concerns raised by Reviewers Bb80, 2jbt, XrhY, and 1jmj. All updates in the revised manuscript are clearly highlighted for ease of review.

---

### **Main Paper & Appendix**

- We have updated ***Section 4.3 Geometry Quality*** to include multi-view reconstruction baselines comparison, real-data geometry metrics and temporal coherence analysis.  ***(To Reviewer Bb80, W1,2,4; Reviewer XrhY, W1, Q2)***
- We have revised **Section 2 (Related Work)** to provide a clearer and more organized discussion of both monocular and multi-view approaches.
- We have added several new experiments in the **Appendix**, including: fairness analysis for monocular baseline comparisons, temporal interpolation capability, grid resolution analysis, computational efficiency evaluation.
- We have added a new **supplementary video** to better illustrate the qualitative results.

We sincerely appreciate the reviewers’ time and constructive feedback, which have substantially improved the clarity and quality of our manuscript.

*Best,*

*Authors*

---

### Author Response · Authors · 2025-12-03
**Subject: Note to AC: Summary of Rebuttal Discussions (Part 1/2)**

**Dear Area Chairs**,

We sincerely appreciate your time and effort in overseeing the review process under these exceptional circumstances. To facilitate your final assessment, we provide a consolidated summary structured as follows: **(1) Summary of Contributions and Reviews**, **(2) Individual Reviewer Assessments**.

---
## 1. Summary of Contributions and Reviews

**Contributions:** Our work introduces a new paradigm for dynamic surface reconstruction based on Parametric Signed Distance Functions (p-SDF), enforcing temporal coherence while handling topological changes, based on three core contributions:

- **Parametric SDF Representation (Reviewer 1jmj, 2jbt, Bb8o, XrhY)**
  - a novel formulation that generalizes static SDF fields into continuous temporal trajectories parameterized by basis functions, naturally enforcing smoothness and continuity.
- **End-to-End Inverse Rendering Framework (Reviewer 1jmj, 2jbt)**
  - which integrates p-SDF with differentiable iso-surfacing and physically-based rendering (PBR) to disentangle geometry, material, and lighting from multi-view video.
- **Robust Dynamic Reconstruction (Reviewer 1jmj, 2jbt, XrhY)**
  - achieving state-of-the-art geometric accuracy and temporal coherence. It robustly handles large-scale non-rigid motions and abrupt topological changes, scenarios where prior deformation-based methods consistently fail.


**Reviews:** The reviews were polarized (**Scores: 8, 8, 4, 2**) yet revealed a consensus on the work's intrinsic merit.

- **Strong Support**: Reviewers **2jbt** and **1jmj** championed the paper, calling p-SDF *"novel"*, *"intuitive"*, and a *"strong contribution"* that is *"simple but effective"*.
- **Common Ground**: Even the critical reviewers (**XrhY** and **Bb8o**) acknowledged the framework is *"well-motivated"* and *"well-formulated"*, recognizing its potential for high-fidelity reconstruction.

**Remaining Concerns & Questions**: As summarized in the table, we have systematically addressed the specific concerns raised by each reviewer during the rebuttal period. Details can be found in the individual responses to each reviewer.

| Concerns & Questions                     | XrhY     | 2jbt    | Bb8o    |  1jmj   |
|------------------------------------------|----------|---------|---------|---------|
| **Real-world Quantitative Metrics**      | ✔        |         |         |         |
| **Comparison with Multi-view Baselines** | ✔        |         | ✔       |         |
| **Efficiency & Grid Resolution**         | ✔        |         |         | ✔       |
| **Temporal Interpolation Validation**    |          | ✔       |         |         |
| **Comparison Fairness (Mono vs Multi)**  |          |         | ✔       |         |
| **Temporal Consistency Improvements**    |          |         | ✔       |         |
| **Topological Changes Capability**       |          |         |         | ✔       |
| -                                        | -        | -       | -       | -       |
| **Initial Scores**                       | 4        | 8       | 2       | 8       |
| **Response Acknowledged**                | -        | -       | -       | ✔       |

We believe we have well resolved these concerns, particularly those requiring clarification for Reviewer Bb8o's key concern about comparison fairness. By clarifying all baselines were retrained with identical multi-view supervision, we proved that their failure stems from architectural limitations rather than data disparity, and established our method's superiority over native multi-view frameworks (NeuS2, AT-GS).

---

> ### Author Response · Authors · 2025-12-03
> **Subject: Note to AC: Summary of Rebuttal Discussions (Part 2/2)**
>
> ## 2. Individual Reviewer Assessments
>
> ### **[Reviewer XrhY] Rating: 4**
>
> - **[W1] Lack of real-world quantitative metrics:**
>
>   Added *Table 3* showing SOTA Chamfer Distance on CMU Panoptic against scanned point clouds.
>
> - **[W2 & W3] Performance on complex geometry (CMU Panoptic):**
>
>   Highlighted *Figure 6* and the *Rebuttal Video* to demonstrate superior topological integrity under severe occlusions compared to broken baselines.
>
> - **[Q1] :Evaluation on human datasets:**
>
>   Clarified that CMU Panoptic covers challenging single and multi-person scenarios. Our SOTA results here (*Table 3*, *Fig. 6*) validate performance on humans.
>
> - **[Q2] Comparison with NeuS2:**
>
>   Added NeuS2 to *Table 2*. Ours outperforms it in PSNR (31.65 vs 30.07) and temporal coherence.
>
> - **[Q3 & Q4] Efficiency & Grid Resolution:**
>
>   Added *Appendix G* (Efficiency Table) and *Table A4* (Resolution Ablation), confirming $96^3$ as the optimal balance.
>
> - **[Q5] Examples under challenging lighting:**
>
>   Highlighted the Cello scene (*Figure 6*) which involves non-Lambertian reflective surfaces. Our PBR modeling successfully recovers geometry where baselines fail.
>
>
> ### **[Reviewer 2jbt] Rating: 8**
>
> - **[Q1] Clarification on evaluation protocol (Spatial vs. Temporal):**
>
>   Confirmed the main results use a standard "Spatial Hold-out" protocol.
>
> - **[Q1] Validation of temporal interpolation:**
>
>   Added *Appendix E* (Temporal Hold-out experiment). Results confirm p-SDF successfully interpolates unseen frames with minimal error, validating its continuous nature.
>
>
> ### **[Reviewer Bb8o] Rating: 2**
> - **[W1] Unfair comparison (Monocular baselines vs. Multi-view method):**
>
>   Clarified that all baselines were retrained with identical multi-view supervision. Added a Controlled Experiment (*Table A6*) proving their failure is due to architectural limits, not data.
>
> - **[W2] Missing Multi-view baselines:**
>
>   Added comparisons with native multi-view methods NeuS2 and AT-GS in *Table 2*, showing our SOTA performance.
>
> - **[W3] Positioning in Related Work (Monocular vs. Multi-view):**
>
>   Thoroughly revised *Section 2* to explicitly categorize methods into "Monocular" and "Multi-view," strictly clarifying our method's positioning to avoid ambiguity.
>
> - **[W4] Temporal consistency improvements:**
>
>   Added *Table 4* (Temporal Coherence Metric) and a *New Video* to quantitatively and qualitatively demonstrate superior stability against baseline flickering.
>
> - **[W5] Claim on "First PBR":**
>
>   Refined the claim and cited SAFT, clarifying our contribution is template-free reconstruction (unlike SAFT).
>
> ### **[Reviewer 1jmj] Rating: 8**
>
> - **[W1 & Q1] Scalability and computational cost:**
>
>   Added *Appendix G*. Our training time (~121 min) is faster than optimization-heavy baselines like SC-GS (253 min). Discussed theoretical scaling and proposed a Sliding Window strategy for long sequences.
>
> - **[W2] Reliance on synchronized multi-view inputs:**
>
>   Acknowledged as a design choice for SOTA geometry.
>
> - **[Q2] Capability to handle topological changes:**
>
>   Added *Appendix A* ("Torus-to-Sphere" experiment). The reviewer acknowledged these results as **"impressive"** and confirmed **"all concerns are addressed"**.
>
> - **[W3] Ablation on basis functions:**
>
>   Added *Table A3*. Results prove the hybrid basis (Polynomial + Fourier) is essential, outperforming "Polynomial Only" by >10 dB.
>
>
> ---
>
> ## Conclusion
> This work presents p-SDF, a principled solution to the long-standing challenge of dynamic surface reconstruction. By parameterizing SDF trajectories, we achieve a breakthrough in temporal coherence and geometric fidelity.
>
> We have responded to address all reviewers' concerns by demonstrating fairness through controlled multi-view experiments, validating performance on real-world humans with quantitative metrics, and proving robustness to topological changes via rigorous ablation. We believe these revisions comprehensively resolve the remaining concerns and firmly establish the validity and significance of our contributions.
>
> Best regards, Authors

---

### Meta-Review · Area_Chair_ZaHC · 2025-12-15

**Summary:**

The reviewers acknowledged the novelty of the parametric SDF proposed in the manuscript and its strong test performance, but expressed concerns primarily in the following aspects:

1.	Clarification of motivation and input/output was needed.

2.	Many comparative methods are designed for monocular video reconstruction, making comparisons with multi-view reconstruction in this paper unfair.

3.	Lack of comparisons on human datasets, such as ZJU-Mocap and PeopleSnapshot.

4.	Handling discontinuous or abrupt spatiotemporal changes.

5.	Missing the task of temporal interpolation.

6.	Quantitative measurement of temporal consistency was required.

7.	Lack of comparison methods, such as NeuS2[Wang et al. 2023], Chen et al. 2025, Zheng et al. 2025.
[Wang et al. 2023] Wang, Yiming, et al. "Neus2: Fast learning of neural implicit surfaces for multi-view reconstruction." Proceedings of the IEEE/CVF International Conference on Computer Vision. 2023.
[Chen et al. 2025] Chen, Decai, et al. "Adaptive and temporally consistent Gaussian surfels for multi-view dynamic reconstruction." 2025 IEEE/CVF Winter Conference on Applications of Computer Vision (WACV). IEEE, 2025.
[Zheng et al. 2025] Zheng, Chengwei, et al. "GauSTAR: Gaussian Surface Tracking and Reconstruction." Proceedings of the Computer Vision and Pattern Recognition Conference. 2025. The reconstruction quality was not evaluated on real-world data.

8.	Performance on examples under challenging lighting conditions.

9.	Limitation on complex scenes and high-detailed surfaces, and grid resolution analysis.

10.	Lack of analysis of training efficiency and memory consumption.

11.	Lack of ablations of the selection of polynomial and Fourier basis functions.

**Reviewer Concerns:**

In response to the reviewers' comments, the authors have provided detailed clarifications for points #1-#4. Additionally, some experiments, evaluations, and analyses corresponding to points #5-#12 have been incorporated into the updated version.

A potentially remaining point is the lack of comparative methods raised by Reviewer XrhY and Reviewer Bb8o (Point #7 above). Although the authors have added numerical experiments of one method (NeuS2) on one dataset (SynthoMotion-360 dataset),  no visualization results were included, the PSNR metric in Table 2 was ignored (mentioned in the discussion), and it was not tested on other datasets, such as the CMU Panoptic Studio dataset. Moreover, the more recent methods [Chen et al. 2025] and [Zheng et al. 2025] suggested by Reviewer Bb8o  were not compared.

 Furthermore, it has been validated in NeuS [Wang et al. 2021] and 2DGS [Huang et al. 2024] that the rendering quality and reconstruction accuracy of NeRF-based and Gaussian-based methods in static scenes surpass those of IDR [Yariv et al. 2020], which utilizes differentiable surface rendering. Therefore, the claim in the paper that the proposed method achieves superior rendering quality and geometric accuracy compared to NeuS2 and Gaussian-based methods raises doubts regarding the validity of this comparison. Reviewer

[Bb8o] also pointed out that some baseline methods were designed for monocular settings, and switching them to a multi-view setup might not have allowed them to achieve their optimal performance.

[Wang et al. 2021] Wang et al. NeuS: Learning Neural Implicit Surfaces by Volume Rendering for Multi-view Reconstruction, 2021.

[Huang et al. 2024] Huang et al. 2D Gaussian Splatting for Geometrically Accurate Radiance Fields, 2024.

[Yariv et al. 2020] Yariv et al. Multiview Neural Surface Reconstruction by Disentangling Geometry and Appearance, 2020.

Another potential limitation is the constrained capability in handling highly detailed surfaces (#9). As evidenced by the supplementary table (Table A7) on memory consumption, the memory footprint of the proposed method significantly exceeds that of existing approaches, thereby restricting its ability to fit high-quality details.

Finally, another aspect overlooked by the reviewers is that since the surfaces are extracted on a per-frame SDF, they lack topological consistency (13s-18s of rebuttal video: the lines under the robot's armpit change frame by frame). This inherently prevents the appearance learning from establishing temporal correspondences. Consequently, the method is unsuitable for monocular video processing. Since the paper utilizes a substantial number of training views (up to 32), it remains unclear whether per-frame multi-view reconstruction could achieve comparable results. The authors did not analyze or compare this variant, which undermines the reliability of the claimed superiority over other methods.

**Reviewer Scores:**

I think that the reviewer whose concerns have been completely resolved (2jbt ( score 8), 1jmj (score 8)) will maintain their original score.

While some of the other reviewers' (XrhY (score 4), Bb8o (score 2))  questions have been addressed, the critical concerns #9 and #7 have not been adequately resolved. Therefore, these two reviewers, especially Bb8o, are likely to maintain their negative scores.

---

### Decision · Program_Chairs · 2026-01-26

Reject